# Saliency is a Possible Red Herring When Diagnosing Poor Generalization

**Joseph D. Viviano**[1,2,*]**, Becks Simpson**[1]**, Francis Dutil**[2]**, Yoshua Bengio**[1,3]**, & Joseph Paul Cohen**[1,†]
[1]Mila, Québec Artificial Intelligence Institute, Université de Montréal
[2]Imagia Cybernetics
[3]CIFAR Senior Fellow
[*]`joseph@viviano.ca`
[†]`joseph@josephpcohen.com`

## Abstract

Poor generalization is one symptom of models that learn to predict target variables using spuriously-correlated image features present only in the training distribution instead of the true image features that denote a class. It is often thought that this can be diagnosed visually using attribution (aka saliency) maps. We study if this assumption is correct. In some prediction tasks, such as for medical images, one may have some images with masks drawn by a human expert, indicating a region of the image containing relevant information to make the prediction. We study multiple methods that take advantage of such auxiliary labels, by training networks to ignore distracting features which may be found outside of the region of interest. This mask information is only used during training and has an impact on generalization accuracy depending on the severity of the shift between the training and test distributions. Surprisingly, while these methods improve generalization performance in the presence of a covariate shift, there is no strong correspondence between the correction of attribution towards the features a human expert has labelled as important and generalization performance. These results suggest that the root cause of poor generalization may not always be spatially defined, and raise questions about the utility of masks as "attribution priors" as well as saliency maps for explainable predictions.

## 1 Introduction

A fundamental challenge when applying deep learning models stems from poor generalization due to covariate shift (Moreno-Torres et al., 2012) when the probably approximately correct (PAC) learning i.i.d. assumption is invalid (Valiant, 1984) i.e. the training distribution is different from the test distribution. One explanation for this is *shortcut learning* or *incorrect feature attribution*, where the model during training overfits to a set of training-data specific decision rules to explain the training data instead of modelling the more general causative factors that generated the data (Goodfellow et al., 2016; Reed & Marks, 1999; Geirhos et al., 2020; Hermann & Lampinen, 2020; Parascandolo et al., 2020; Arjovsky et al., 2019; Zhang et al., 2016).

In medical imaging, poor generalization due to test-set distribution shifts are common and this problem is exacerbated by small cohorts. Previous work has hypothesized that this poor generalization is in part due to the presence of confounding variables in the training data such as acquisition site or other image acquisition parameters because attribution maps (aka saliency maps; Simonyan et al. (2014)) produced by the trained model do not highlight features that a human expert would use to make a diagnosis (Zech et al., 2018; DeGrave et al., 2020; Badgeley et al., 2019; Zhao et al., 2019; Young et al., 2019). Previous researchers have made the assumption that saliency maps can demonstrate that the model is not overfit or behaving unexpectedly (Pasa et al., 2019; Tschandl et al., 2020). We started this work under the same assumption only to find the contradictory evidence we present in this paper. In this work, we set out to test the hypothesis that models with good generalization properties have attribution maps which only utilize the class-discriminative features to make predictions, by explicitly regularizing the models to ignore confounders using *attribution priors* (Erion et al., 2019; Ross et al., 2017), i.e., to make predictions using the correct anatomy (as a doctor would). We evaluated whether this regularization would A) improve out of distribution generalization, and B) change feature attribution to be more like the attribution priors. If there exists a relationship between the attribution map and generalization performance, we would expect these

regularizations to positively impact both generalization and attribution quality simultaneously. To evaluate attribution quality, we define *good attribution* to be an attribution map that agrees strongly with expert knowledge in the form of a binary mask on the input data.

We show that the existing and proposed feature-attribution-aware methods help facilitate generalization in the presence of a train-test distribution shift. However, while feature-attribution-aware methods change the attribution maps relative to baseline, there is no strong correlation between generalization performance and good attribution. This in turn challenges the assumption made in previous works that the "incorrect" attribution maps were indicative of poor generalization performance. This suggests that A) efforts to validate model correctness using attribution maps may not be reliable, and B) that efforts to control feature attribution using masks on the input may not function as expected. All code and datasets for this paper are publicly available[1]. Our contributions include:

- A synthetic dataset that encourages models to overfit to an easy to represent confounder instead of a more complicated counting task.
- Two new tasks constructed from open medical datasets which have a correlation between the pathology and either *imaging site* (site pathology correlation; SPC) or *view* (view pathology correlation; VPC), and we manipulate the nature of this correlation differently in the training and test distributions to create a distribution shift (Figure 5), introducing confounding variables as observed in previous work (Zhao et al., 2019; DeGrave et al., 2020).
- Evaluation of existing methods for controlling feature attribution using mask information; *right for the right reasons* (*RRR*; Ross et al. (2017)), *GradMask* (Simpson et al., 2019), and adversarial domain invariance (Tzeng et al., 2017; Ganin & Lempitsky, 2015).
- A new method for controlling feature attribution based on minimizing activation differences between masked and unmasked images (*ActDiff*).
- Evaluate the relationship between generalization improvement and feature attribution in real-life out of distribution generalization tasks with traditional classifiers.

## 2 RELATED WORK

It is a well-documented phenomenon that convolutional neural networks (CNNs), instead of building object-level representations of the input data, tend to find convenient surface-level statistics in the training data that are predictive of class (Jo & Bengio, 2017). Previous work has attempted to reduce the model's proclivity to use confounding features by randomly masking out regions of the input (DeVries & Taylor, 2017), forcing the network to learn representations that aren't dependent on a single input feature. However, this regularization approach gives no control over the kinds of representations learned by the model, so we do not include it in our study.

Recently, multiple approaches have proposed to control feature representations by penalizing the model for producing saliency gradients outside of a regions of interest indicating the class-discriminative feature (Simpson et al., 2019; Zhuang et al., 2019; Rieger et al., 2019). These approaches were introduced by *Right for the Right Reasons* (*RRR*), which showed impressive improvements in attribution correctness on synthetic data (Ross et al., 2017). The follow-up work has generally demonstrated a small improvement in generalization accuracy on real data, and much more impressive results on synthetic data. Another feature attribution-aware regularization approach additionally dealt with class imbalances by increasing the impact of the gradients inside the region of interest of the under-represented class Zhuang et al. (2019).

One alternative to saliency-based methods, which can be noisy due to the ReLU activations allowing irrelevant features to pass through the activation function (Kim et al., 2019), would be to leverage methods that aim to produce *domain invariant* features in the latent space of the network. These methods regularize the network such that the latent representations of two or more "domains" are encouraged to be as similar as possible, often by minimizing a distance metric or by employing an adversary that is trained to distinguish between the different domains (Kouw & Loog, 2019; Ganin & Lempitsky, 2015; Tzeng et al., 2015; Liu & Tuzel, 2016). In this work, we view the masked version of the input as the training domain and the unmasked version of the input as the test domain, and compare these approaches with saliency-based approaches for the task of reducing the model's reliance on confounding features. To the best of our knowledge, these strategies have not been tried to control feature attribution.

---

[1] https://github.com/josephdviviano/saliency-red-herring

## 3 METHODS

**Domain Invariance with the Activation Difference and Adversarial Loss:** Leveraging ideas from domain adaptation, we introduce two methods designed to make the model invariant to features arising from outside of the *attribution priors*. The first is the embarrassingly simple *Activation Difference* (*ActDiff*) approach, which simply penalizes, for each input, the $L2$-normed distance between the masked and unmasked input's latent representations. This loss is most similar to a style transfer approach which transforms a random noise image into one containing the same layer-wise activations as some target image (Gatys et al., 2015) to apply a visual aesthetic to some input semantic contents, and encourages the network to build features which appear inside the masked regions even though it always sees the full image during training. Therefore we minimize

$$\mathcal{L}_{act} = \sum_{(\mathbf{X}_{masked}, \mathbf{x}) \in D} \mathcal{L}_{clf} + \lambda_{act} ||o_l(\mathbf{x}_{masked}) - o_l(\mathbf{x})||_2, \qquad (1)$$

where $o_l(\mathbf{x})$ are the pre-activation outputs for layer $l$ of the $n$-layer encoder $f(x)$ when the network is presented with the original data $\mathbf{x}$, $o_l(\mathbf{x}_{masked})$ are the pre-activations outputs for layer $l$ when presented with masked data $\mathbf{x}_{masked}$, and $\mathcal{L}_{clf}$ is the standard cross entropy loss. We define $\mathbf{x}_{masked} = \mathbf{x} \cdot \mathbf{x}_{seg} + \text{shuffle}(\mathbf{x}) \cdot (1 - \mathbf{x}_{seg})$ where background pixels are shuffled uniquely for each presentation to the network in order to destroy any spatial information available in those regions of the image without introducing any consistent artefacts into the image or shifts in the distribution of input intensities.

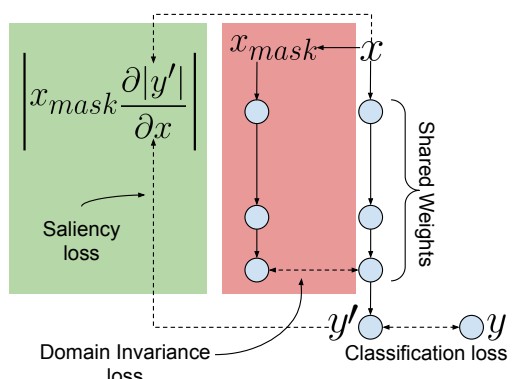

This method is at a high level similar to using maximum mean discrepancy (MMD) for domain adaptation (Baktashmotlagh et al., 2016; 2013) where instead of minimizing the distance between the means of the two domain distributions, we instead minimize the distance between the features directly. One could replace the $L2$ norm with any $Lk$ norm, and choices of $k < 2$ might be useful when regularizing larger

Figure 1: Schematic of the model used in all experiments. The backbone is an 18-layer ResNet. The *domain invariance* penalties, *ActDiff* and *Adversarial*, are applied to the linear layer after average pooling (although they could be applied at any location in the network). In contrast, the *saliency* penalties, *GradMask* and *Right for the Right Reasons*, are applied on the input space. All losses are denoted using standard dashed lines.

latent representations as $L2$ distances collapse to a constant value in extremely high dimensional spaces due to the curse of dimensionality (Aggarwal et al., 2001). We found during experimentation that regularizing the pre-activations led to better results at the cost of longer time to convergence, perhaps because the $L2$ norm is more effective when the feature vectors are not sparse, but we leave this conjecture to future work.

The second approach we explore employs a discriminator $\mathcal{D}(\cdot)$ optimized to distinguish between latent representations arising from passing $\mathbf{x}_{masked}$ or $\mathbf{x}$ through the encoder $f(\cdot)$ (Tzeng et al., 2017; Goodfellow et al., 2014). Simultaneously, we optimize the encoder $f(\cdot)$ to fool the discriminator and still produce representations that are predictive of the output class. Therefore, $\mathcal{D}$ and $f$ are optimized using the $\mathcal{L}_{\mathcal{D}}$ and $\mathcal{L}_f$ respectively:

$$\mathcal{L}_{\mathcal{D}} = \lambda_{disc}\big(\mathbb{E}_{\mathbf{x}_{masked}}[log\mathcal{D}(f(\mathbf{x}_{masked}))] + \mathbb{E}_{\mathbf{x}_{masked}}[log(1 - \mathcal{D}(f(\mathbf{x})))]\big) \qquad (2)$$

$$\mathcal{L}_f = \mathcal{L}_{clf} + \lambda_{disc}\big(\mathbb{E}_{\mathbf{x}}[log(1 - \mathcal{D}(f(\mathbf{x}_{masked})))] + \mathbb{E}_{\mathbf{x}}[log\mathcal{D}(f(\mathbf{x}))]\big) \qquad (3)$$

This approach is similar to the one employed in (Janizek et al., 2020), where the authors encouraged the model to be invariant to the view of the X-Ray (Lateral vs PA), and here 'view' denotes whether the image is masked or not. $\mathcal{D}(\cdot)$ had three fully-connected hidden layers of size 1024 before outputting a binary prediction. To facilitate the stability of training, we updated $\mathcal{D}(\cdot)$ more frequently than the encoder treating this ratio as a hyperparameter. We optimized $\mathcal{D}(\cdot)$ independently using Adam with a distinct learning rate, and applied spectral normalization to the hidden layers of $\mathcal{D}(\cdot)$.

**Direct Attribution Control with the Right for the Right Reasons and GradMask Loss:** These input gradient attribution regularizers directly control which regions of the input are desirable for

determining the class label by penalizing saliency outside of a defined input mask. The most basic gradient based "input feature attribution" can be calculated as $\frac{\partial |\hat{y}_i|}{\partial \mathbf{x}}$ for each input $x$ (Simonyan et al., 2014; Lo et al., 2015). In the binary classification case, *RRR* (Ross et al., 2017) calculates saliency of the summed *log probabilities* of the 2 output classes with respect to the input $\mathbf{x}$,

$$\mathcal{L}_{rrr} = \sum_{(\mathbf{x}_{seg}, \mathbf{x}) \in D} \mathcal{L}_{clf} + \lambda_{rrr} \cdot \frac{\partial \left( log(\hat{p_0}) + log(\hat{p_1}) \right)^2}{\partial \mathbf{x}} \cdot (1 - \mathbf{x}_{seg}), \tag{4}$$

where $(1 - \mathbf{x}_{seg})$ is a binary mask that covers everything outside the defined regions of interest. The numerator in the RRR loss can be extended to the multi-class case by substituting $\sum_k^K log(\hat{p_k})$. *GradMask* (Simpson et al., 2019) similarly calculates saliency using the contrast between the two output logits, and is only defined for the binary classification case,

$$\mathcal{L}_{gradmask} = \sum_{(\mathbf{x}_{seg}, \mathbf{x}) \in D} \mathcal{L}_{clf} + \lambda_{gradmask} \cdot \left| \frac{\partial |\hat{y}_0 - \hat{y}_1|}{\partial \mathbf{x}} \cdot (1 - \mathbf{x}_{seg}) \right|_2, \tag{5}$$

where $\hat{y}_0$ and $\hat{y}_1$ are the predicted outputs for our two classes.

**Classify Masked:** We evaluated the effect imposing the attribution prior by simply training a model using masked data (and evaluating it using unmasked data) as a control experiment. The data was masked by shuffling the pixels outside of the mask during training, as was done for the domain invariance experiments to produce $\mathbf{x}_{masked}$. We refer to these experiments as *Masked*.

## 4 EXPERIMENTS

**Feature Attribution Analysis and Visualization:** We evaluated our ability to control feature attribution using three methods. We first calculated the *input gradient* as the absolute gradient $G$ of the input with respect to the prediction made for all images of the positive class $G = |\frac{\partial \hat{y}_1}{\partial \mathbf{x}}|$ (Simonyan et al., 2014; Ancona et al., 2018). We also calculated *integrated gradients*, which produces attribution maps that are invariant to the specific model trained (or function $f(\cdot)$) and are sensitive to all input features that drive the prediction (Sundararajan et al., 2017). This is done by integrating the gradients along the straightline path between the input image $x_i$ and an all-zero baseline image $x_i'$, $IG_x ::= (x_i - x_i') \times \int_{\alpha=0}^{1} \frac{\partial f(x' + \alpha \times (x - x'))}{\partial x_i} \partial \alpha$. The integral was approximated over 200 steps for each image using GaussLegendre quadrature. We finally evaluated a non gradient-based attribution method *occlusion* where we divided the input image into a $15 \times 15$ grid, and for each element of the grid, recorded the magnitude of the change to the model's output logits if that region is removed from the input (by setting these regions to zero) (Zeiler & Fergus, 2014). For the two gradient-based attribution methods, we smoothed the resulting saliency map with a small Gaussian kernel ($\sigma = 1$) to remove pixel-wise variance in the attribution values that amount to noise. The attribution maps were then carried forward to calculate a binarized version such that the top $p$ percentile attribution values were set to 1, where $p$ is dynamically set to the number of pixels in the mask that it is being compared to. These binarized values were used to calculate the localization accuracy between binarized attribution map and the ground-truth segmentation using the intersection over union (IoU; $I_u(A, B) = \frac{A \cap B}{A \cup B}$) for all images in the test sets for all models trained. To aid in visualization, we also thresholded these attribution maps arbitrarily at the $50^{th}$ percentile so that the most attributed regions of the image are easily seen when overlaid on the anatomy. We present visualizations of the results for all three methods in the Appendix. These methods were implemented using the Captum library (Kokhlikyan et al., 2020).

**Model, Optimization, Hyperparameters, and Search Parameters:** The backbone of all experiments was the ResNet-18 model from Torchvision (He et al., 2016; Marcel & Rodriguez, 2010). For medical dataloaders we use the TorchXRayVision library (Cohen et al., 2020b;a). To see an overview of how the losses described above relate to the architecture, see Figure 1. All models were trained using the Adam optimizer (Kingma & Ba, 2014) using early stopping on the validation loss.

Hyperparamters were selected for all models using a Bayesian hyperparameter search with a single seed, 5 random initializations, 20 iterations, and trained for a maximum of 100 epochs with a patience of 20. Final models were trained using the best hyperparameters found for 100 epochs on 10 seeds.The learning rate for all models was searched on a log uniform scale between $[10^{-5} \; 10^{-2}]$. The *ActDiff*, *Adversarial*, *GradMask*, and *RRR* lambdas were all searched between $[10^{-4} \; 10]$, each

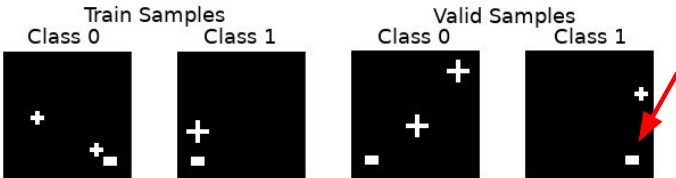

Figure 2: Example images from $D_{train}$ and $D_{valid}$ from both classes. In both distributions, cross size can vary between samples. In $D_{train}$, two crosses (denoting class 0) are always accompanied by a box $x_c$ in the bottom right-hand corner, while a single cross (denoting class 1) is always accompanied by a confounder in the bottom left-hand corner. In $D_{valid}$, the relationship between classes and crosses remains the same, but the logic governing the location of the counfounder is reversed. The confounder is indicated with a red arrow.

on a log uniform scale. For *Adversarial*, we searched for the optimal discriminator : encoder training ratio of $[2 : 1 \ 10 : 1]$, and the discriminator learning rate was searched on a log uniform scale between $[10^{-4} \ 10^{-2}]$. To retain context around the masked region in the Synthetic dataset, we dilated the mask by applying a Gaussian blur with a hyperparameter $\sigma$ to the mask and then binarize the result; $\sigma$ was searched on a uniform scale between $[0 \ 2]$. For all experiments, the batch size was 16, weight of the classification cross entropy loss was 1. In performing our hyperparameter searches, we found that certain algorithms were much easier to tune than others. We present the final best validation AUCs encountered for all iterations for all experiments in Appendix Figure A.1 and Table 3.

## 4.1 SYNTHETIC DATASET

**Protocol:** Consider a classification problem where there exists some confounder feature $x_c$ in the data (a vector of variables $x$) that is perfectly correlated with one of the output classes $y$ in the training distribution $D_{train}$ such that $p(y = 1|x_c) = 1$, while in the validation distribution $D_{valid}$, $p(y = 0|x_c) = 1$ (Figure 2). In this scenario, predicting using $x_c$ is easier than predicting using the true features that would denote class membership and a classifier trained on $D_{train}$ with traditional classification loss would predict the incorrect class with 100% probability on $D_{valid}$.

We generated a dataset with these conditions where class membership is denoted by the number of crosses present in the image (1 vs. 2 crosses) that can appear in any position, but there exists in all $D_{train}$ images a confounder that always appears in the same position and is perfectly correlated with $y$ (Figure 2). The logic governing the position of the confounder was inverted for $D_{valid}$. This dataset had 500 training, 128 validation, and 128 test examples respectively. Segmentations were generated for all crosses in the dataset to facilitate all feature-attribution controlling losses. In cases where the model relies on the confounder to make a class prediction, we expected 0.0 AUC for the validation and test sets.

**Results:** The mean and standard deviation of the validation AUC over all 10 models for the 100 epochs of training for the best-performing hyperparameters are shown in Figure 3, showing the variance in performance over model initializations and data splits. We demonstrate the effect of controlling feature attribution by showing the mean integrated gradient map calculated for 500 positive examples in the test set in Figure 4.

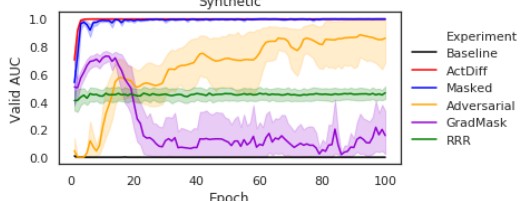

Figure 3: Synthetic Dataset Valid AUC over 100 epochs, averaged over 10 seeds.

The validation curves (Figure 3), demonstrate that the baseline model quickly learns the shortcut in the training set and shows below-chance generalization (AUC=0). The *Masked* approach is never given the opportunity to build a representation of the confounder and is therefore able to generalize while still giving attribution to the confounder. The domain invariance approaches, *ActDiff* and *Adversarial* also learn the task, although convergence is much more difficult for the *Adversarial* approach. The two saliency-based feature-attribution regularizers, *GradMask*, and *RRR*, show unstable training and never reach optimal performance, although they do prevent the model from overfitting to the confounder. The saliency maps for these models (Figure 4) show that all feature attribution controlling approaches (C-F) successfully refocus the model's attention away from the confounder, unlike the baseline and *Masked* models (A-B), leading to improved generalization. This strong relationship between IoU and AUC for this dataset can be observed for all seeds in Figure 8. As we will see, this relationship does not hold for real world data.

| Image | A: Baseline | B: Masked | C: Adversarial | D: ActDiff | E: GradMask | F: RRR |
|-------|-------------|-----------|----------------|------------|-------------|--------|

Synthetic: IG

Figure 4: The average integrated gradients (IG) saliency maps from 500 randomly-selected $D_{test}$ images for the Synthetic dataset. The first image shows the mean of the samples, with the mean of the masks superimposed in red. The remainder of the panels show the mean of the gradients after taking the absolute value, smoothing, and thresholding at the $50^{th}$ percentile.

## 4.2 X-Ray Dataset with Site-Pathology Correlation

**Protocol:** We introduced a covariate-shift into a joint dataset of X-Rays drawn from two different imaging centres: the PadChest (Bustos et al., 2019) sample and the NIH Chestx-Ray8 (Wang et al., 2017) sample. In Figure 5, we can see examples of the distribution shift between imaging site (top) and imaging view (bottom; used in the following section). In the case that the prevalence of disease is in any way correlated with either of these confounders, the model is likely to learn to use these differences to make a prediction regardless of their medical relevance (rightmost column). In this dataset, models were required to predict the presence of Cardiomegaly (enlarged heart). A site-driven overfitting signal has previously been reported when combining these datasets (Zech et al., 2018), where the model often attributes importance to the patient's shoulder when making a prediction. We also observed site-driven differences in regions far from the lungs (see the mean X-Ray from each dataset in Figure 5), and therefore hypothesized we could improve overfitting performance by masking out the edges of the image using a circular mask. To induce the covariate shift, we preferentially sampled the positive class from one of the two imaging centres in the training set to produce a site-pathology correlation (SPC), and then produce validation and test set where the reverse relationship is true.

In the training set, 90% of the unhealthy patients were drawn from the PadChest dataset and the remaining 10% of the unhealthy patients were drawn from the NIH dataset, and the reverse logic was followed for the validation and test sets. In all splits the classes and site distributions were always balanced, making it tempting for the classifier to use a site-specific feature when predicting the class in the presence of site-pathology correlation ($N_{train} = 5542$, $N_{valid} = 2770$, $N_{test} = 2770$). We also constructed a version of this dataset where the classes are drawn equally from both sites for the training, validation, and test sets, i.e., a dataset where there exists no site pathology correlation (No SPC). All images were $224 \times 224$ pixels.

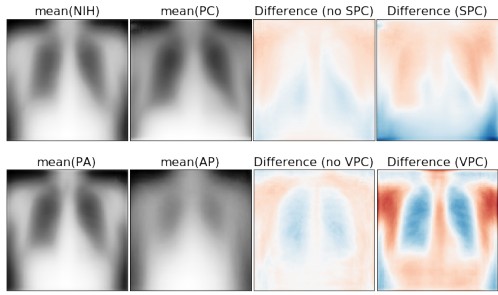

Figure 5: Illustration of the effect of Site-Pathology Correlations (SPC; top row) and View-Pathology Correlations (VPC; bottom row). The first two columns contain the mean of all X-Rays collected from a particular site/view. The third column shows the class-conditional mean difference between images taken from sick and healthy patients when pathology is uncorrelated with site/view, and the fourth column shows the same when pathology is correlated with site/view.

**Results:** We trained each model on both the SPC (90%) and No SPC datasets (Table 1). A ResNet trained on the No SPC dataset scores a test AUC of $0.93 \pm 0.01$, while one trained in the presence of a strong SPC scores a test AUC of $0.70 \pm 0.05$, indicating poorer generalization under a SPC. Validation curves across seeds show that model optimization was much easier and more consistent for all models when trained on the No SPC dataset (Appendix Figure A.2). The *ActDiff*, *Masked*, and *GradMask* approaches only improved classification performance over the baseline model in the presence of an SPC (*GradMask* performed only slightly better than baseline for the No SPC dataset). For *ActDiff* and *GradMask*, an improvement in classification performance was associated with an improvement in attribution (measured as an improvement in the IoU between the binarized saliency map and the masks), but this pattern was not observed for the masked experiment, even though the *Masked* experiment produced superior generalization (it is worth noting that the *Masked* performance was more variable across seeds). Furthermore, the *GradMask* approach improved attribution in the No SPC case, but this was not associated with a meaningful improvement in generalization performance. Figure 6 shows the mean saliency map calculated from a subset of all test images from all 10 models. Differences in the baseline model's attribution (A) in the SPC and No SPC case are apparent: in particular, the SPC

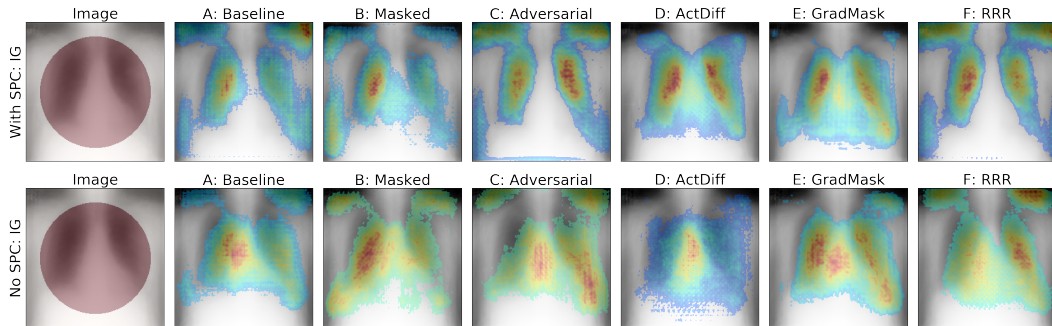

Figure 6: Mean integrated gradients (IG) saliency maps from 500 randomly-selected $D_{test}$ images in the X-Ray dataset with a site-pathology correlation (SPC; top), and without (No SPC; bottom) after taking the absolute value, smoothing, and thresholding at the $50^{th}$ percentile across all 10 seeds. The first image shows the mean input image of the model over all examples shown, with the mask overlaid in red.

| | Experiment | AUC | IoU Input Grad | IoU Integrated | IoU Occlude |
|---|---|---|---|---|---|
| | | | **With SPC** | | |
| Baseline | Classification | $0.70 \pm 0.05$ | $0.40 \pm 0.03$ | $0.33 \pm 0.03$ | $0.41 \pm 0.04$ |
| | Masked | $\mathbf{0.77 \pm 0.09}$ | $0.38 \pm 0.04$ | $0.38 \pm 0.05$ | $0.38 \pm 0.04$ |
| Domain Invariance | ActDiff | $\mathbf{0.73 \pm 0.03}$ | $\mathbf{0.62 \pm 0.04}$ | $\mathbf{0.58 \pm 0.04}$ | $\mathbf{0.62 \pm 0.02}$ |
| | Adversarial | $0.68 \pm 0.07$ | $0.36 \pm 0.02$ | $0.32 \pm 0.02$ | $0.40 \pm 0.04$ |
| Saliency Based | GradMask | $\mathbf{0.74 \pm 0.03}$ | $\mathbf{0.57 \pm 0.02}$ | $\mathbf{0.50 \pm 0.04}$ | $\mathbf{0.50 \pm 0.04}$ |
| | RRR | $0.57 \pm 0.09$ | $0.38 \pm 0.02$ | $0.33 \pm 0.03$ | $0.43 \pm 0.04$ |
| | Experiment | AUC | IoU Input Grad | IoU Integrated | IoU Occlude |
| | | | **No SPC** | | |
| Baseline | Classification | $0.93 \pm 0.01$ | $0.48 \pm 0.03$ | $0.42 \pm 0.03$ | $0.49 \pm 0.02$ |
| | Masked | $0.90 \pm 0.01$ | $0.42 \pm 0.02$ | $0.39 \pm 0.03$ | $0.46 \pm 0.03$ |
| Domain Invariance | ActDiff | $0.63 \pm 0.20$ | $0.21 \pm 0.25$ | $0.26 \pm 0.24$ | $0.21 \pm 0.26$ |
| | Adversarial | $0.92 \pm 0.01$ | $0.44 \pm 0.03$ | $0.40 \pm 0.04$ | $0.46 \pm 0.03$ |
| Saliency Based | GradMask | $0.94 \pm 0.00$ | $\mathbf{0.55 \pm 0.02}$ | $\mathbf{0.48 \pm 0.03}$ | $\mathbf{0.53 \pm 0.02}$ |
| | RRR | $0.93 \pm 0.01$ | $0.46 \pm 0.03$ | $0.42 \pm 0.04$ | $0.48 \pm 0.04$ |

Table 1: X-Ray Cardiomegaly test results for the best valid epoch over 100 epochs on the X-Ray dataset with and without a site-pathology correlation (SPC). Results averaged over 10 seeds with the standard deviation. Bold indicates better than Classification baseline and chance. IoU scores were calculated on 100 randomly-selected test-set images for each seed (total of 1000).

model shows a greater reliance on the shoulder for producing a prediction. The *Masked* experiment (B) exacerbates this behavior but is accompanied by improved generalization with higher variance across seeds. In contrast, *ActDiff* and *GradMask* appear to refocus the saliency away from the shoulder (D-E), reflected in the increased IoU scores for both the *ActDiff* and *GradMask* approaches for the SPC data. The weak relationship between IoU and AUC can be observed for all seeds in Figure 8. These results provide evidence that controlling feature attribution only facilitates generalization in the presence of a covariate shift, but improves saliency in both the presence or absence of a covariate shift, implying a weak relationship between good attribution and good generalization.

### 4.3 X-RAY DATASET WITH VIEW-PATHOLOGY CORRELATION

**Protocol:** Here we made use of the RSNA Pneumonia challenge (Shih et al., 2019) dataset, which includes images taken from antero-posterior (AP) and posterior-anterior (PA) views, and where the task is to predict pneumonia. We introduced a covariate shift into this dataset by sampling the positive and negative classes during training such that $90\%$ of the positive classes are sampled from one view in the training set and the opposite view in the validation and test sets. We call this a view-pathology correlation (VPC). After drawing the samples from both views and balancing the classes, the data in each split was as follows: $N_{train} = 2696$, $N_{valid} = 1348$, $N_{test} = 1348$. All images with Pneumonia have one or more bounding boxes indicating the predictive regions. In Figure 5 we can see the strong effect of the VPC, such that the lungs, shoulders, and torso show strong interclass differences in the VPC case. We also constructed a No VPC dataset in the same way as during the previous experiments, and all images were $224 \times 224$ pixels.

**Results:** Similarly to the SPC case, baseline AUC in the presence of VPC ($0.20 \pm 0.03$) is substantially lower than the No VPC baseline ($0.76 \pm 0.02$), and additionally far below chance performance ($0.50$), illustrating the strong covariate shift in this dataset. Appendix Figure A.2 shows that valida-

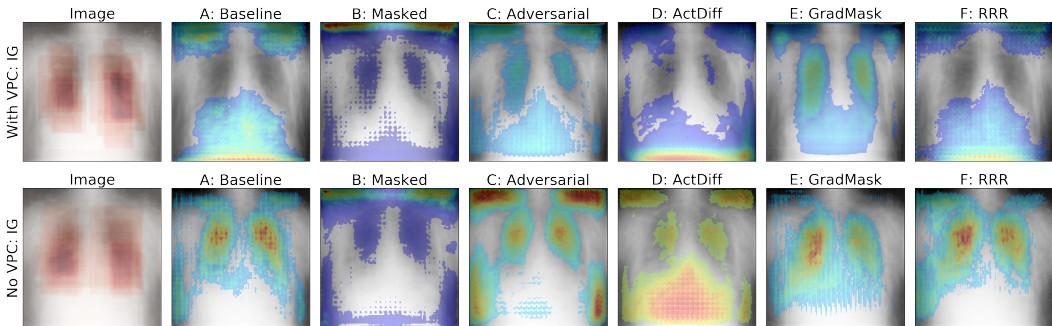

Figure 7: Mean integrated gradients (IG) saliency maps from 500 randomly-selected $D_{test}$ images in the RSNA dataset with a view-pathology correlation (VPC; top), and without (No VPC; bottom) after taking the absolute value, smoothing, and thresholding at the $50^{th}$ percentile across all 10 seeds. The first image shows the mean input image of the model over all examples shown, with the mask overlaid in red.

| | | | With VPC | | |
| --- | --- | --- | --- | --- | --- |
| | Experiment | AUC | IoU Input Grad | IoU Integrated | IoU Occlude |
| Baseline | Classification | $0.20 \pm 0.03$ | $0.10 \pm 0.02$ | $0.05 \pm 0.02$ | $0.08 \pm 0.02$ |
| | Masked | $0.56 \pm 0.16$ | $0.07 \pm 0.03$ | $0.08 \pm 0.02$ | $0.04 \pm 0.01$ |
| Domain Invariance | ActDiff | $\mathbf{0.68 \pm 0.05}$ | $0.08 \pm 0.01$ | $0.02 \pm 0.00$ | $0.04 \pm 0.01$ |
| | Adversarial | $\mathbf{0.62 \pm 0.10}$ | $0.10 \pm 0.05$ | $0.05 \pm 0.03$ | $0.07 \pm 0.04$ |
| Saliency Based | GradMask | $0.49 \pm 0.06$ | $\mathbf{0.17 \pm 0.06}$ | $\mathbf{0.08 \pm 0.02}$ | $0.08 \pm 0.03$ |
| | RRR | $0.21 \pm 0.04$ | $\mathbf{0.11 \pm 0.02}$ | $0.05 \pm 0.01$ | $0.08 \pm 0.02$ |
| | | | No VPC | | |
| | Experiment | AUC | IoU Input Grad | IoU Integrated | IoU Occlude |
| Baseline | Classification | $0.76 \pm 0.02$ | $0.17 \pm 0.02$ | $0.10 \pm 0.02$ | $0.11 \pm 0.01$ |
| | Masked | $0.50 \pm 0.02$ | $0.07 \pm 0.03$ | $0.08 \pm 0.02$ | $0.04 \pm 0.02$ |
| Domain Invariance | ActDiff | $0.60 \pm 0.02$ | $0.13 \pm 0.01$ | $0.05 \pm 0.01$ | $0.06 \pm 0.01$ |
| | Adversarial | $0.65 \pm 0.02$ | $0.09 \pm 0.02$ | $0.04 \pm 0.01$ | $0.07 \pm 0.01$ |
| Saliency Based | GradMask | $0.75 \pm 0.01$ | $0.16 \pm 0.03$ | $0.11 \pm 0.02$ | $0.11 \pm 0.01$ |
| | RRR | $0.75 \pm 0.01$ | $0.16 \pm 0.04$ | $0.10 \pm 0.02$ | $0.12 \pm 0.02$ |

Table 2: RSNA Pneumonia test results for the best valid epoch over 100 epochs on the RSNA dataset with and without a view-pathology correlation (VPC). Results averaged over 10 seeds with the standard deviation. Bold indicates better than both the Classification baseline and chance. IoU scores were calculated on 100 randomly-selected test-set images for each seed (total of 1000).

tion performance was more variable in the presence of VPC between models, and that performance was generally lower. The *ActDiff*, *Masked*, and *Adversarial* approaches improved classification performance over the baseline model and chance performance in the presence of VPC. For only the *GradMask* model was an improvement in classification performance associated with an improvement in saliency, and the *Masked* experiment produced IoU scores substantially worse than baseline (again, *Masked* performance was more variable across seeds). No model improved attribution or generalization performance over baseline in the No VPC case. Figure 7 shows the mean saliency map calculated from a subset of all test images from all 10 models. Differences in the *Baseline* model's saliency (A) in the VPC and No VPC case are apparent: in particular, the VPC model shows a greater reliance on the abdomen and shoulders for producing a prediction. The *Masked* experiment (B) again exacerbates this behavior but is accompanied by improved generalization over baseline. The *Adversarial*, *ActDiff*, and *GradMask* models (C-E) appear to refocus the attribution toward the lungs for for the VPC case, but the mean *Adversarial* IoU score dropped over the full dataset, and *ActDiff* resulted in a minor IoU score increase. The *GradMask* approach produced the best attribution as measured by IoU but does not produce a model that outperforms random guessing on the test set. Again, these results suggest that the use of attribution priors can have a substantial effect on generalization performance in the absence of meaningful changes to the saliency towards what would be expected from the attribution priors, and vice versa. The weak relationship between IoU and AUC can be observed for all seeds in Figure 8.

## 5 LIMITATIONS AND CONCLUSIONS

**Limitations:** Our results show that models explicitly regularized to not use features constructed from outside of the mask still attribute features outside of the mask at test time. This is likely due to

the fact that convolutional models are still able to construct discriminative filters from inside of the mask which might share properties with features found outside of the mask. The model is free to use any aspect of the input image to make a prediction during test time. In particular, the methods we explored are not guaranteed to negate any confounding variables that exist within the mask, which could explain these results, and might additionally give rise to underspecified trained models, as evidenced by the models with high test set variability across seeds (D'Amour et al., 2020). This is particularly true for the *Masked* models, for which the representations constructed from inside of the masks were not regularized in any way. Methods that address this underspecification would make for valuable future work. Furthermore, while the results suggest that attribution maps are a misleading indicator of generalization performance, we only evaluated these algorithms using small sample sizes with large covariate shifts between the training and test distributions in a medical context. Future work should more thoroughly evaluate the relationship between generalization performance and feature attribution in other settings before we can draw strong conclusions about the non-existence of this relationship. Furthermore, methods which control feature attribution without using masks for enforcing domain knowledge might show a stronger relationship between IoU and AUC, and future work should explore this possibility. These methods would also be more general, as input masks are only available in specific domains and can be expensive to obtain.

**Conclusions:** We hypothesized that poor generalization performance could be partially attributable to classifiers exploiting spatially-distinct confounding features (or shortcuts) as have been previously diagnosed using saliency maps (Badgeley et al., 2019; Zech et al., 2018). We evaluated the performance of multiple mitigation strategies on a synthetic dataset and two real-world datasets exhibiting covariate shift, using attribution priors in the form of a segmentation of the discriminative features.

In our synthetic dataset, we defined a spatially-distinct confounder that prevented generalization for standard classifiers, and demonstrated that imposing an attribution prior would facilitate generalization and improve attribution maps. In real data, we found that attribution priors also facilitate generalization when there exists a covariate shift between the training and testing distribution, but these methods hurt generalization when there was not a covariate shift. Surprisingly, improvement in generalization performance was not reliably accompanied by improvement in attribution: this is particularly apparent for the *ActDiff* model on the RSNA VPC dataset, which had a meaningful positive impact on AUC and a very negative impact on attribution. Figure 8 summarizes the relationship between generalization (AUC) and feature attribution (IoU) across all seeds for all experiments with a covariate shift: the positive correlation between AUC and IoU apparent for the synthetic data was not present in the real world datasets.

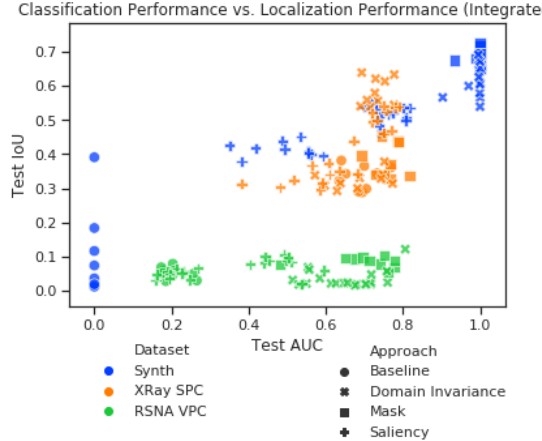

Figure 8: Test AUC vs. Test IoU (good saliency) for all seeds evaluated with covariate shift. Models presented grouped by "Approaches": baseline, input masking, domain-invariance, and saliency-based approaches. All localizations were computed using integrated gradients, see Appendix Figure A.13 for all attribution methods.

In summary, we find a tenuous relationship between good saliency and generalization performance in real datasets. Many models that exhibit good generalization performance do not obtain good attribution, and vice-versa. While our methods exert influence on the features the model constructs from the data, in real data we found no evidence of a relationship between improved generalization and improved feature attribution. We now doubt the validity of using saliency maps for diagnosing whether a model is overfit because improving generalization using attribution priors is not accompanied by improved attribution.

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

## ACKNOWLEDGMENTS

We thank Julia Vetter and Pascal Vincent for insightful discussions. This work is partially funded by a grant from the Fonds de Recherche en Santé du Québec and the Institut de valorisation des données (IVADO). This work utilized the supercomputing facilities managed by Compute Canada and Calcul Quebec. We thank AcademicTorrents.com for making data available for our research.

# A   APPENDIX

## A.1   ARCHITECTURE OF THE RESNET-18 MODEL

The backbone of the model used for all experiments was the 18-layer resnet available from torchvision (Marcel & Rodriguez, 2010) with a small architecture adjustment near the input of the network. The official model's input convolutional layer had a kernel size of 7, stride of 2, and padding of 3. In our experiments, we found this larger kernel size on the inputs led to less specific saliency maps (data not shown), and we therefore changed the kernel size to 3, stride to 1, and padding to 1 for this layer. Immediately following this layer, the official model employed a maxpooling operation. We found this operation to produce unusable noisy saliency maps, so we also disabled this operation for all experiments.

## A.2   HYPERPARAMETER SENSITIVITY

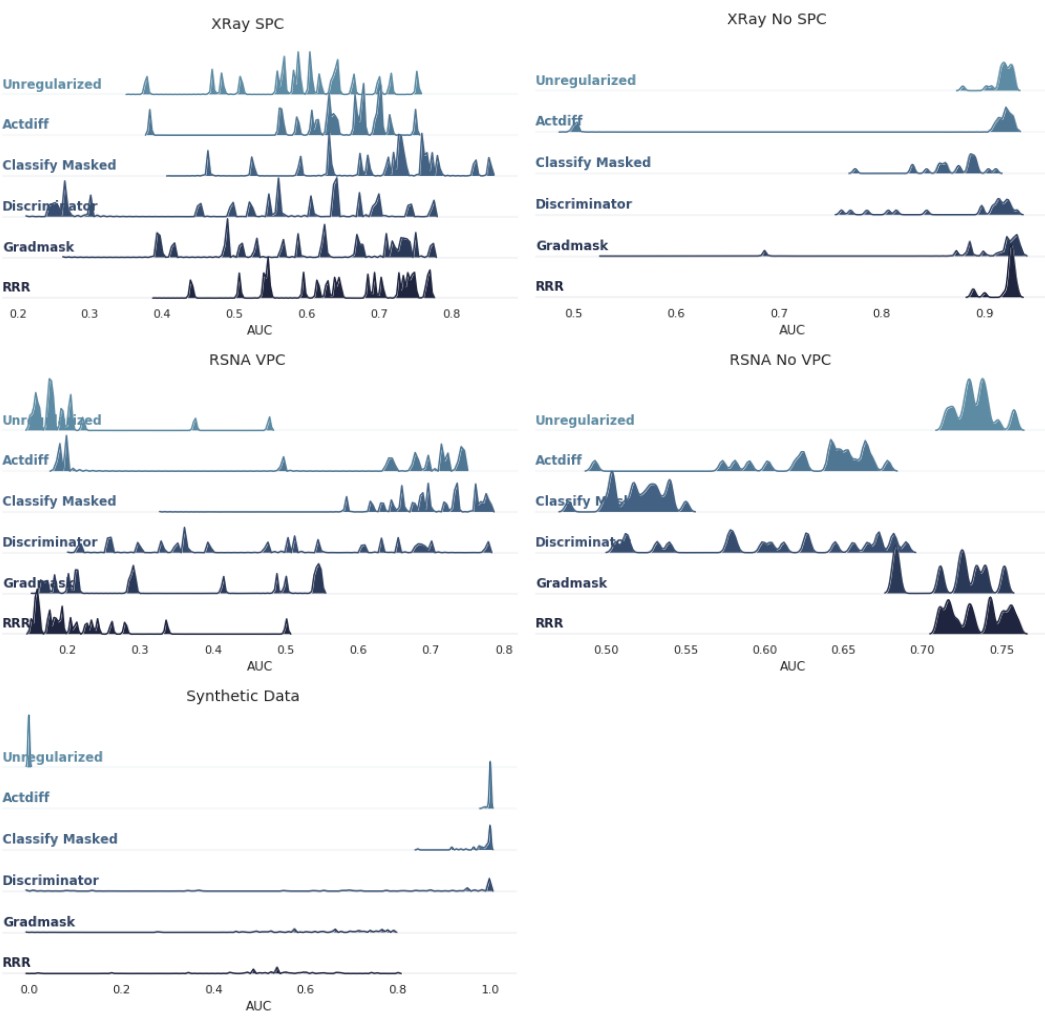

Figure A.1: The best validation AUC for all hyperparameter settings tested during the model tuning. On the synthetic dataset, domain-adaptation approaches were able to solve the task, as did the model that simply observed masked versions of the data. The saliency-based approaches were generally more sensitive to hyperparameters. In the presence of a covariate shift (SPC / VPC), hyperparameter tuning was more difficult on all real-world data.

### A.3 VALIDATION CURVES FOR THE REAL WORLD DATASETS

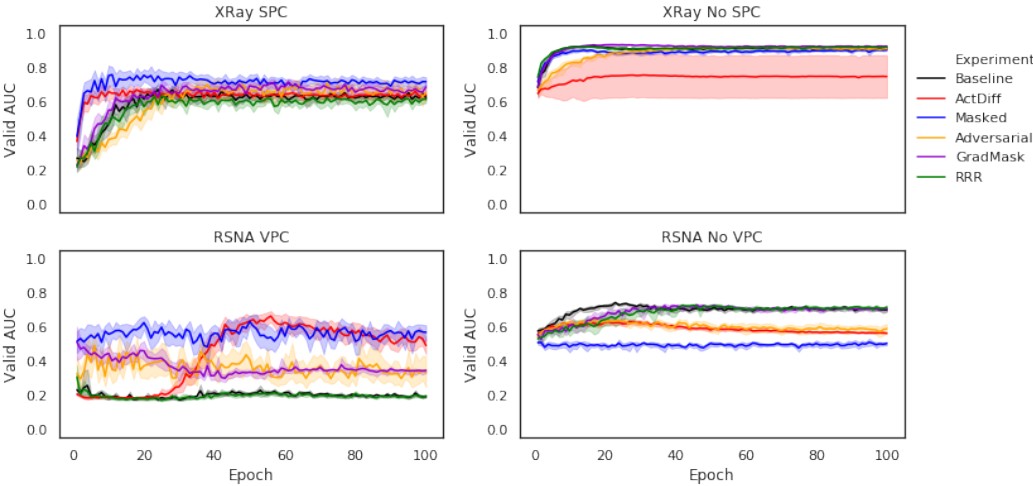

Figure A.2: All real dataset (X-Ray and RSNA) valid AUC across the 100 epochs of training. Mean and standard deviation presented over 10 seeds.

### A.4 BEST HYPERPARAMETERS FOR ALL EXPERIMENTS

| Dataset | Experiment | Learning Rate | Lambda | Disc Iteration Ratio | Disc Learning Rate |
|---|---|---|---|---|---|
| Synthetic | Classification | $3.11 \times 10^{-4}$ | – | – | – |
| | Masked | $3 \times 10^{-5}$ | – | – | – |
| | Actdiff | $3.11 \times 10^{-4}$ | $1.15 \times 10^{-1}$ | – | – |
| | Adversarial | $3.71 \times 10^{-4}$ | 10 | 10:1 | 0.01 |
| | GradMask | $1 \times 10^{-2}$ | $2.03 \times 10^{-1}$ | – | – |
| | RRR | $1 \times 10^{-5}$ | 7.56 | – | – |
| X-Ray SPC | Classification | $9.99 \times 10^{-3}$ | – | – | – |
| | Masked | $1 \times 10^{-2}$ | – | – | – |
| | Actdiff | $5.38 \times 10^{-4}$ | $1 \times 10^{-1}$ | – | – |
| | Adversarial | $9.83 \times 10^{-4}$ | $1.93 \times 10^{-2}$ | 2:1 | $1 \times 10^{-2}$ |
| | GradMask | $9.24 \times 10^{-3}$ | $3.85 \times 10^{-1}$ | – | – |
| | RRR | $7.98 \times 10^{-4}$ | $1 \times 10^{-4}$ | – | – |
| X-Ray No SPC | Classification | $1 \times 10^{-2}$ | – | – | – |
| | Masked | $6.87 \times 10^{-4}$ | – | – | – |
| | Actdiff | $8.92 \times 10^{-3}$ | $5.78 \times 10^{-1}$ | – | – |
| | Adversarial | $3.82 \times 10^{-3}$ | $5.66 \times 10^{-4}$ | 4:1 | $4.27 \times 10^{-3}$ |
| | GradMask | $1 \times 10^{-2}$ | $9.8 \times 10^{-2}$ | – | – |
| | RRR | $2.35 \times 10^{-4}$ | 9.57 | – | – |
| RSNA VPC | Classification | $9.99 \times 10^{-3}$ | – | – | – |
| | Masked | $2.8 \times 10^{-3}$ | – | – | – |
| | Actdiff | $4.1 \times 10^{-5}$ | 1.6 | – | – |
| | Adversarial | $9.83 \times 10^{-4}$ | 10 | 10:1 | $1.16 \times 10^{-4}$ |
| | GradMask | $1 \times 10^{-2}$ | 6.57 | – | – |
| | RRR | $9.99 \times 10^{-3}$ | $1 \times 10^{-4}$ | – | – |
| RSNA No VPC | Classification | $5.5 \times 10^{-4}$ | – | – | – |
| | Masked | $2.84 \times 10^{-3}$ | – | – | – |
| | Actdiff | $1 \times 10^{-4}$ | 3.88 | – | – |
| | Adversarial | $3.4 \times 10^{-5}$ | $1 \times 10^{-4}$ | 6:1 | $2.35 \times 10^{-4}$ |
| | GradMask | $1 \times 10^{-2}$ | $1 \times 10^{-4}$ | – | – |
| | RRR | $3.11 \times 10^{-4}$ | $2.3 \times 10^{-1}$ | – | – |

Table 3: Best hyperparameters found using for each hyperparameter search. Discriminator iteration ratios presented as the update ratios of the discriminator : encoder.

## A.5 Visualizations of Input Gradients, Integrated Gradients, and Occlusion-based Attribution Maps

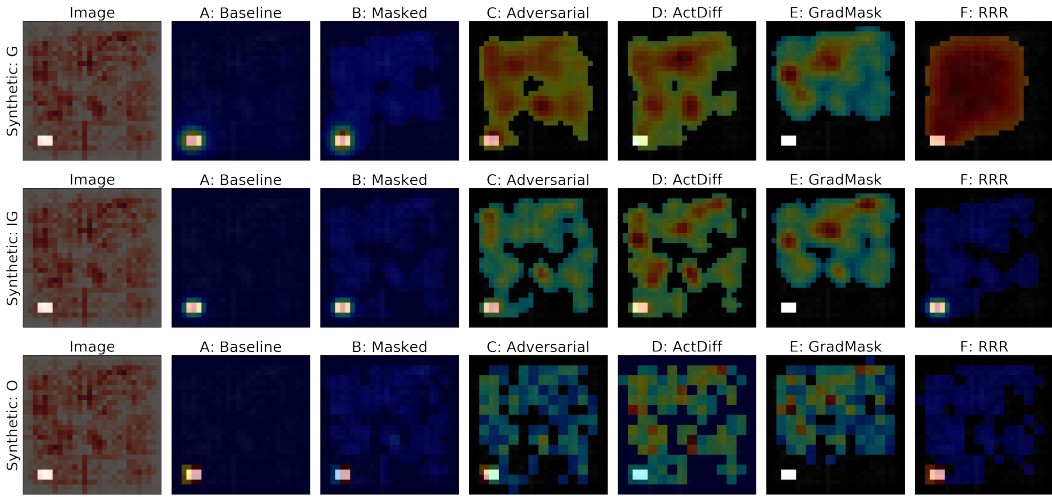

Figure A.3: Mean input gradients (G; top), integrated gradients (IG; middle), and occlusion-based (O; bottom) saliency maps from 500 randomly-selected $D_{test}$ images in the Synthetic dataset after taking the absolute value, smoothing, and thresholding at the $50^{th}$ percentile across all 10 seeds. The first image shows the mean input image of the model over all examples shown, with the mask overlaid in red.

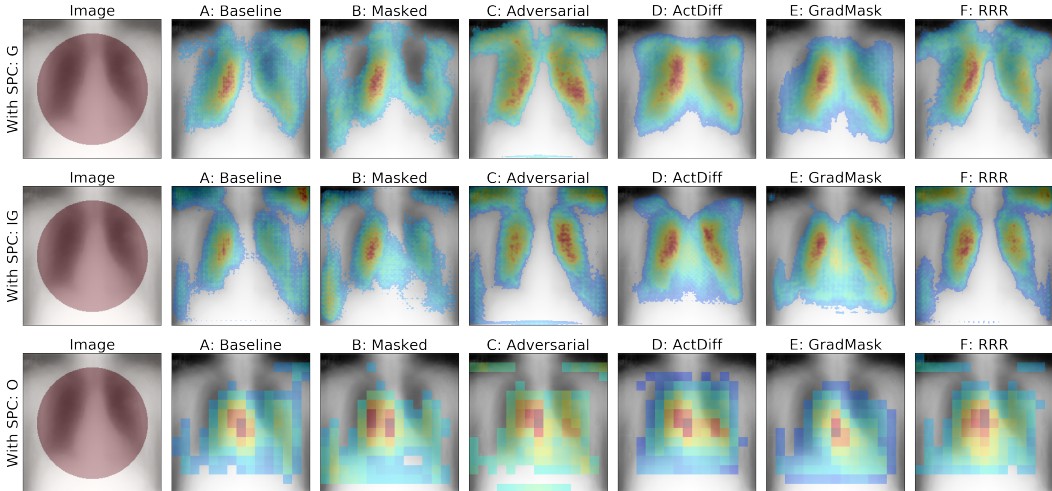

Figure A.4: Mean input gradients (G; top), integrated gradients (IG; middle), and occlusion-based (O; bottom) saliency maps from 500 randomly-selected $D_{test}$ images in the X-Ray dataset with a site-pathology correlation (SPC) after taking the absolute value, smoothing, and thresholding at the $50^{th}$ percentile across all 10 seeds. The first image shows the mean input image of the model over all examples shown, with the mask overlaid in red.

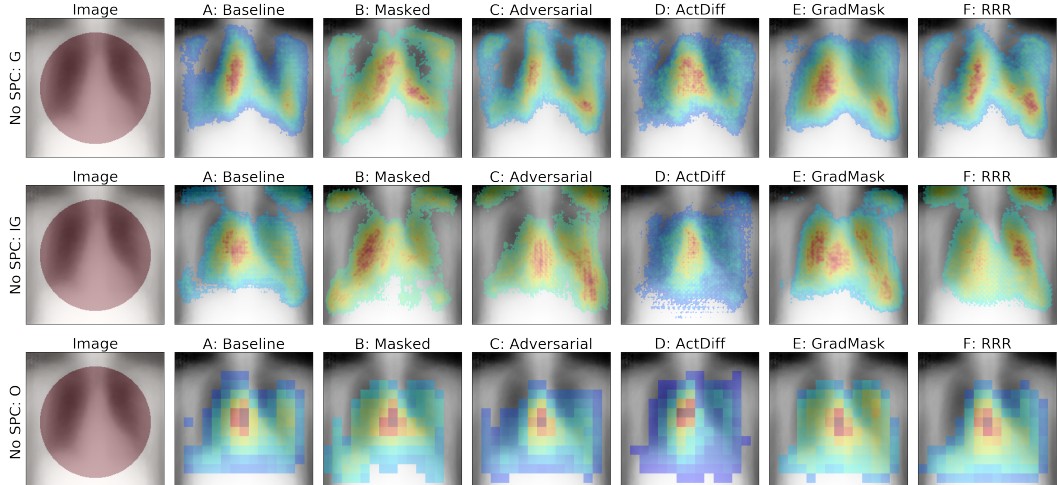

Figure A.5: Mean input gradients (G; top), integrated gradients (IG; middle), and occlusion-based (O; bottom) saliency maps from 500 randomly-selected $D_{test}$ images in the X-Ray dataset without a site-pathology correlation (No SPC) after taking the absolute value, smoothing, and thresholding at the $50^{th}$ percentile across all 10 seeds. The first image shows the mean input image of the model over all examples shown, with the mask overlaid in red.

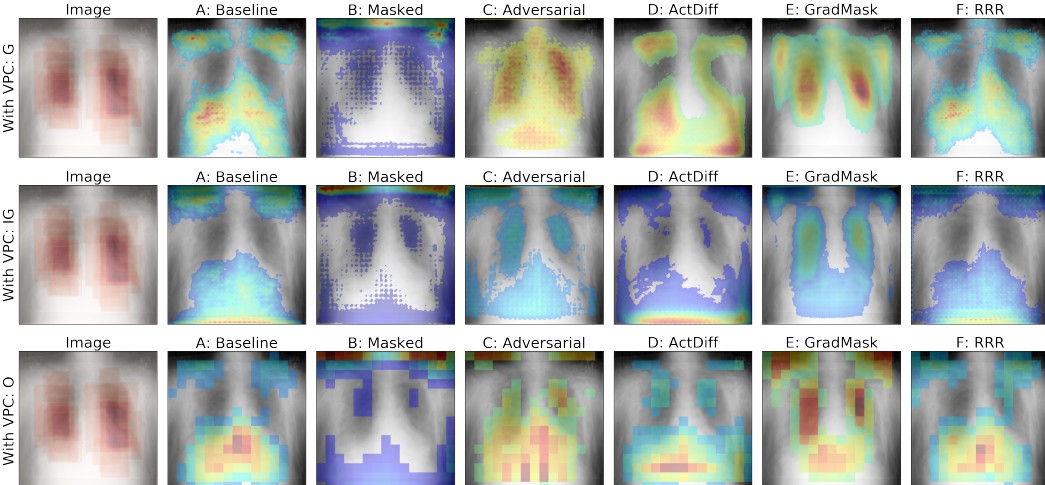

Figure A.6: Mean input gradients (G; top), integrated gradients (IG; middle), and occlusion-based (O; bottom) saliency maps from 500 randomly-selected $D_{test}$ images in the RSNA dataset with a view-pathology correlation (VPC) after taking the absolute value, smoothing, and thresholding at the $50^{th}$ percentile across all 10 seeds. The first image shows the mean input image of the model over all examples shown, with the mask overlaid in red.

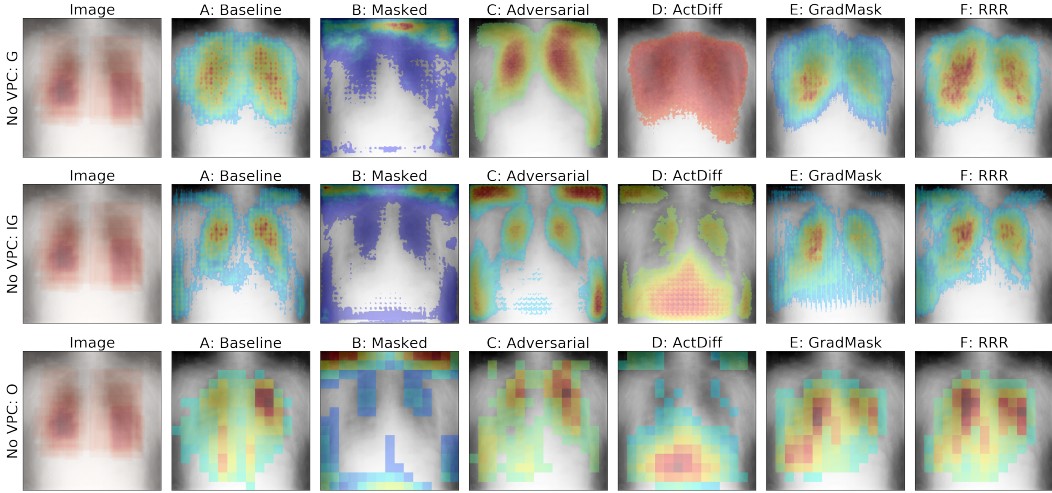

Figure A.7: Mean input gradients (G; top), integrated gradients (IG; middle), and occlusion-based (O; bottom) saliency maps from 500 randomly-selected $D_{test}$ images in the RSNA dataset with a no view-pathology correlation (No VPC) after taking the absolute value, smoothing, and thresholding at the $50^{th}$ percentile across all 10 seeds. The first image shows the mean input image of the model over all examples shown, with the mask overlaid in red.

## A.6 SALIENCY MAPS FOR INCORRECT AND CORRECT PREDICTIONS

Here, we visualize the mean gradients from samples the model gets correct (top) vs. incorrect (bottom) for 100 samples from each seed for a total of 1000 samples. While the synthetic dataset shows obvious differences for correct and incorrect samples, no such pattern is obvious for the real world datasets.

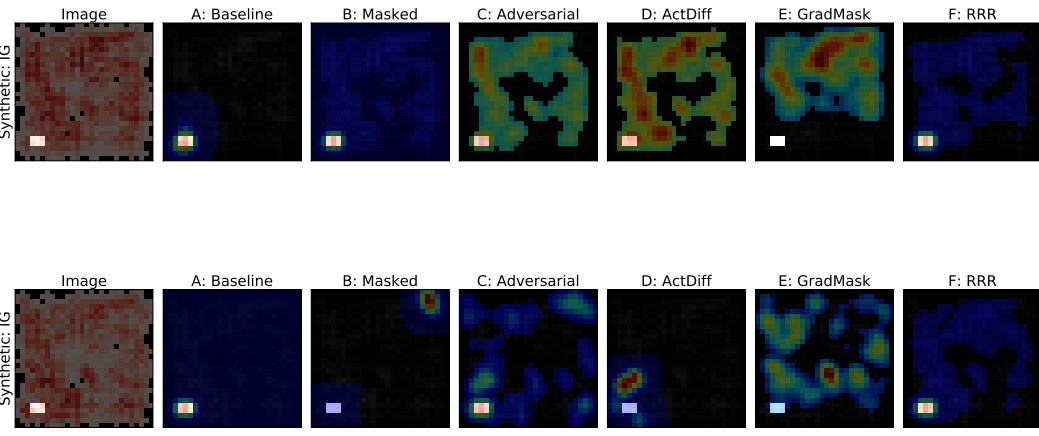

Figure A.8: Mean integrated gradients attribution maps from 500 randomly-selected $D_{test}$ images in the synthetic dataset, showing correct predictions (top) separately from incorrect predictions (bottom), after taking the absolute value, smoothing, and thresholding at the $50^{th}$ percentile across all 10 seeds. The first image shows the mean input image of the model over all examples shown, with the mask overlaid in red.

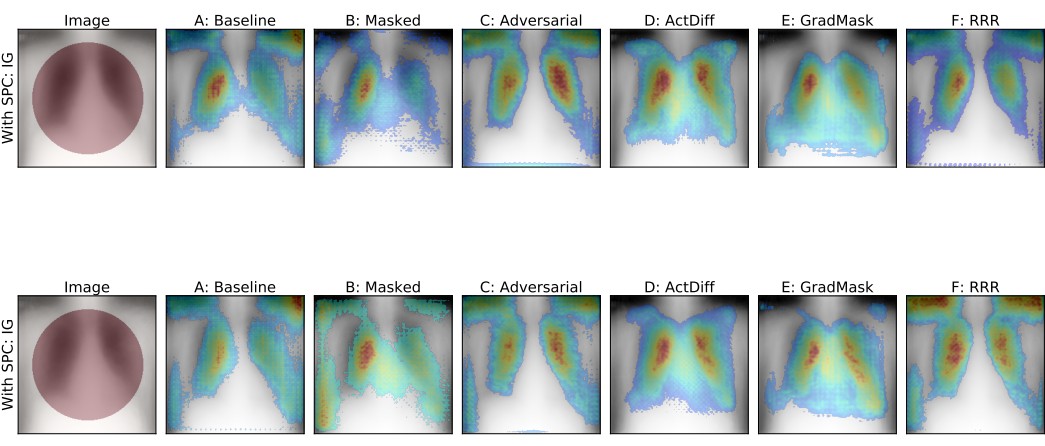

Figure A.9: Mean integrated gradients attribution maps from 500 randomly-selected $D_{test}$ images in the X-Ray SPC dataset, showing correct predictions (top) separately from incorrect predictions (bottom), after taking the absolute value, smoothing, and thresholding at the $50^{th}$ percentile across all 10 seeds. The first image shows the mean input image of the model over all examples shown, with the mask overlaid in red.

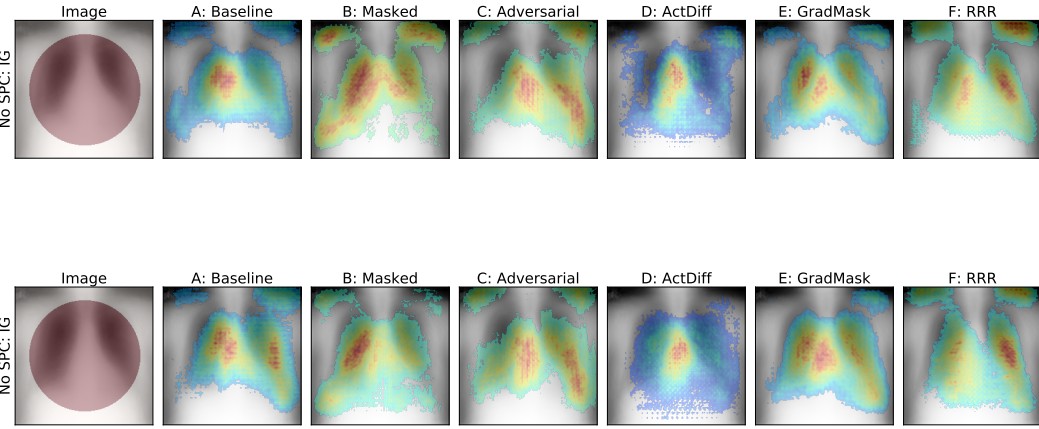

Figure A.10: Mean integrated gradients attribution maps from 500 randomly-selected $D_{test}$ images in the X-Ray No SPC dataset, showing correct predictions (top) separately from incorrect predictions (bottom), after taking the absolute value, smoothing, and thresholding at the $50^{th}$ percentile across all 10 seeds. The first image shows the mean input image of the model over all examples shown, with the mask overlaid in red.

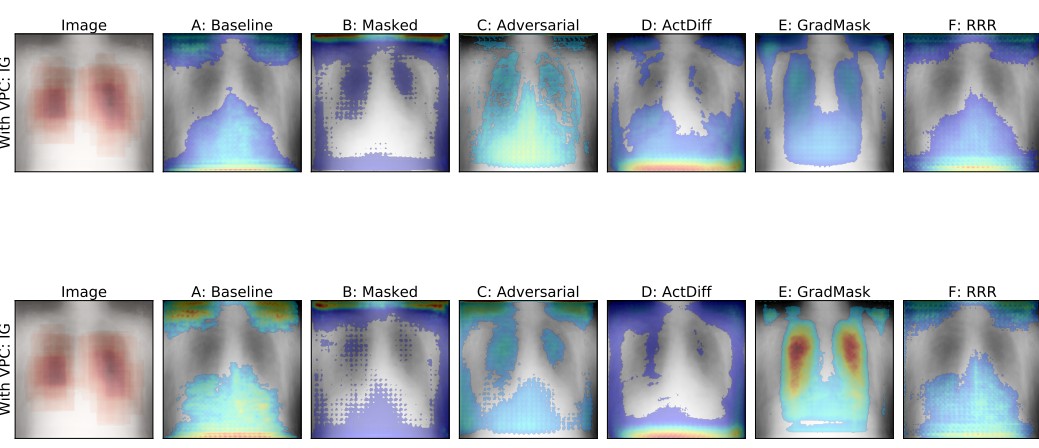

Figure A.11: Mean integrated gradients attribution maps from 500 randomly-selected $D_{test}$ images in the RSNA VPC dataset, showing correct predictions (top) separately from incorrect predictions (bottom), after taking the absolute value, smoothing, and thresholding at the $50^{th}$ percentile across all 10 seeds. The first image shows the mean input image of the model over all examples shown, with the mask overlaid in red.

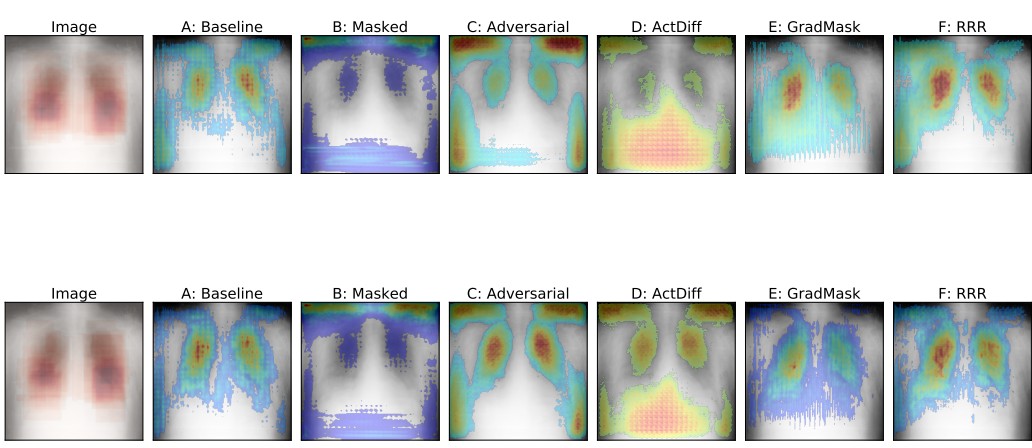

Figure A.12: Mean integrated gradients attribution maps from 500 randomly-selected $D_{test}$ images in the RSNA No VPC dataset, showing correct predictions (top) separately from incorrect predictions (bottom), after taking the absolute value, smoothing, and thresholding at the $50^{th}$ percentile across all 10 seeds. The first image shows the mean input image of the model over all examples shown, with the mask overlaid in red.

## A.7 Generalization Ability vs Attribution Quality for All Attribution Methods

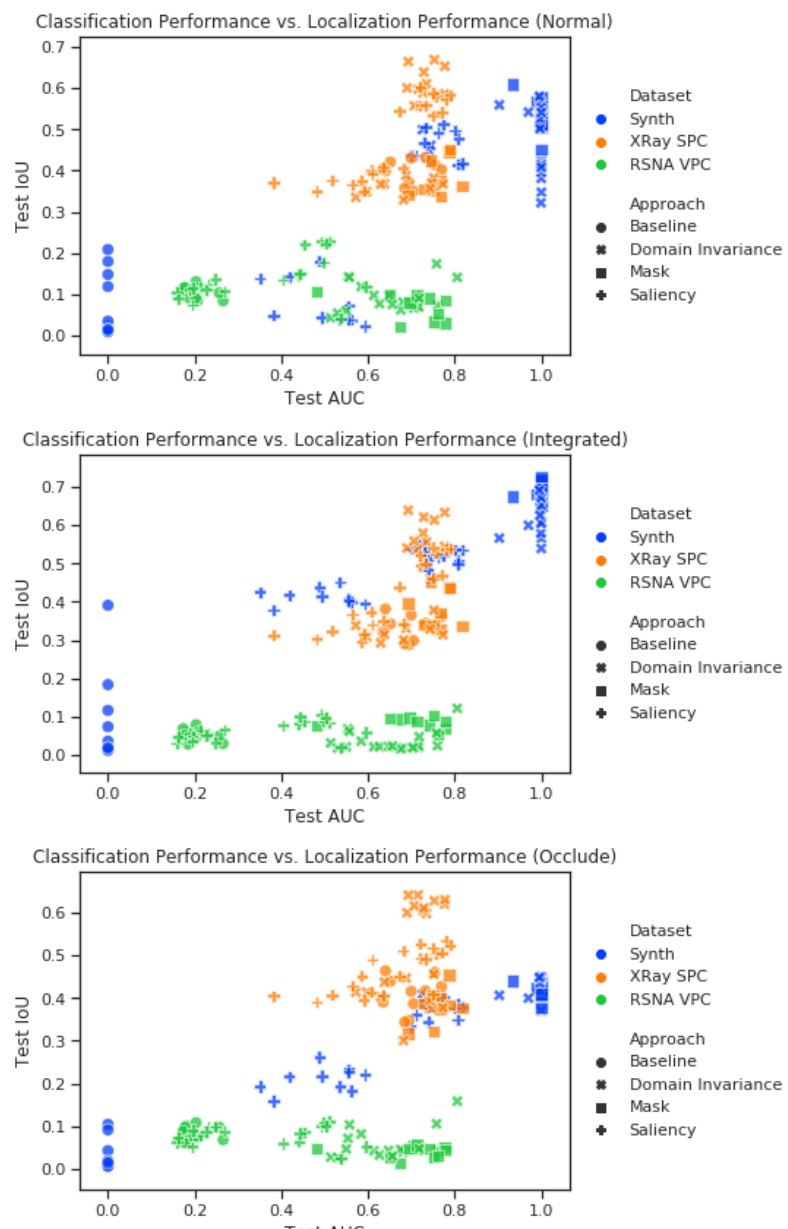

Figure A.13: Test AUC vs. Test IoU (good saliency) for all seeds evaluated with covariate shift. Models presented grouped by "Approaches": baseline, input masking, domain-invariance, and saliency-based approaches. We present the relationship between classification performance and localization performance using input gradients (top left), integrated gradients (top right) and occlusion methods (bottom) for comparison.

## A.8 Actdiff Lambda Evaluation

In real-world experiments without a covariate shift, we found the ActDiff method performed worse than baseline. We suspected this was due to the minimum lambda value in the range considered during hyperparameter tuning ($[1 \times 10^{-4} 10]$ being too large: the ActDiff model should perform equivalently to the baseline if this lambda value is small enough. We tested this on both datasets by running a new hyperparamater search with an expanded range for the actdiff lambda $[1 \times 10^{-16} 10]$

which we call the wide search, and report the test set values over 10 seeds in Table 4. While we recovered the baseline performance for the No SPC dataset, we were unable to do so on the No VPC dataset. We suspect that the ActDiff penalty is quite strong in the RSNA data due to the smaller input masks, making even very small lambda values of ActDiff a powerful regularizer of the model, and making the hyperparameter search difficult given a fixed compute budget. We conclude that the difficulty of tuning the hyperparamaters in situations where no covariate shift exists in the data is a potential downside of this approach.

| Dataset | Experiment | Search | Learning Rate | Lambda | AUC |
|---------|-----------|--------|---------------|--------|-----|
| X-Ray | SPC | Baseline | – | – | $0.70 \pm 0.05$ |
| | | Normal | $5.38 \times 10^{-4}$ | $1 \times 10^{-1}$ | $0.73 \pm 0.03$ |
| | | Wide | $5.64 \times 10^{-4}$ | $1 \times 10^{-16}$ | $0.71 \pm 0.04$ |
| | No SPC | Baseline | – | – | $0.93 \pm 0.01$ |
| | | Normal | $8.92 \times 10^{-3}$ | $5.78 \times 10^{-1}$ | $0.63 \pm 0.20$ |
| | | Wide | $4.77 \times 10^{-4}$ | $1 \times 10^{-16}$ | $0.93 \pm 0.01$ |
| RSNA | VPC | Baseline | – | – | $0.20 \pm 0.03$ |
| | | Normal | $4.1 \times 10^{-5}$ | 1.6 | $0.68 \pm 0.05$ |
| | | Wide | $1.7 \times 10^{-5}$ | $9.26 \times 10^{-13}$ | $0.71 \pm 0.04$ |
| | No VPC | Baseline | – | – | $0.76 \pm 0.02$ |
| | | Normal | $1 \times 10^{-4}$ | 3.88 | $0.60 \pm 0.02$ |
| | | Wide | $1 \times 10^{-5}$ | $1.7 \times 10^{-15}$ | $0.61 \pm 0.02$ |

Table 4: Hyperparamaters and Test AUC (mean $\pm$ standard deviation) for the actdiff models with a wide hyperparameter search range on the X-Ray SPC and RSNA VPC datasets across 10 seeds.

## A.9 INDIVIDUAL SAMPLES

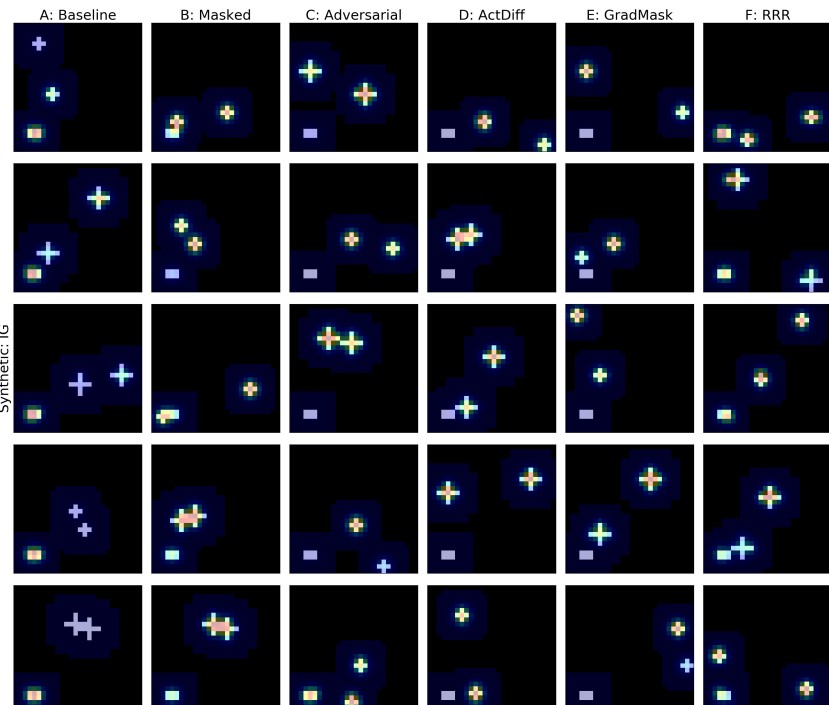

Figure A.14: Randomly-selected images for the synthetic dataset from 5 trained models (1 per row) for each training method, computed using integrated gradients.

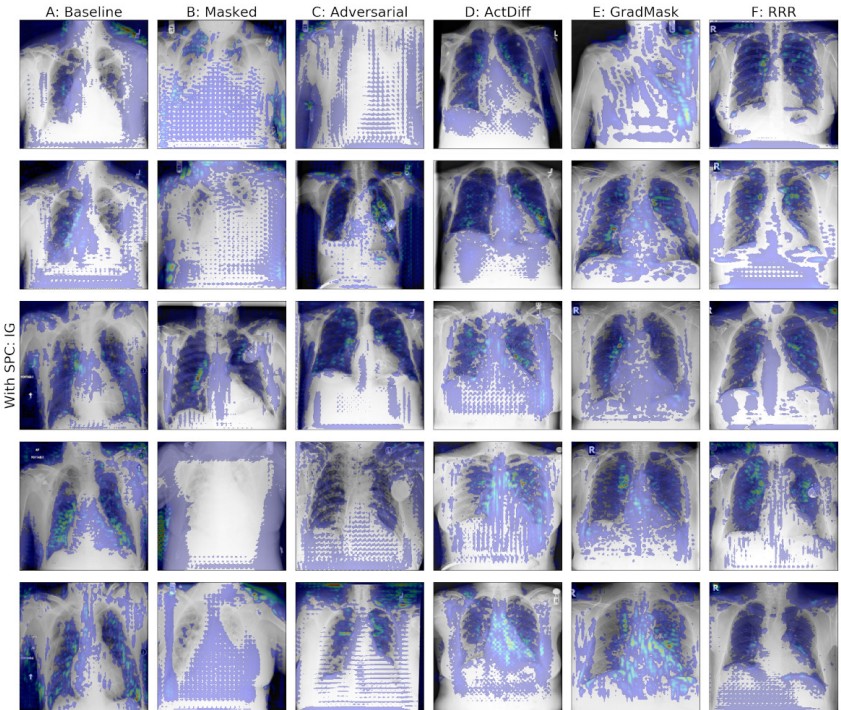

Figure A.15: Randomly-selected images for the X-Ray SPC dataset from 5 trained models (1 per row) for each training method, computed using integrated gradients.

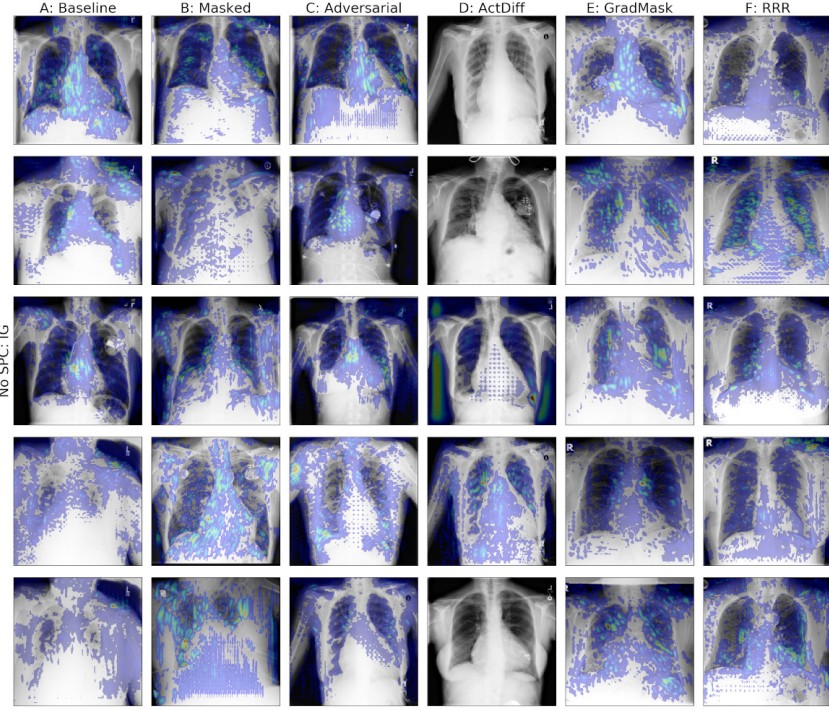

Figure A.16: Randomly-selected images for the X-Ray No SPC dataset from 5 trained models (1 per row) for each training method, computed using integrated gradients.

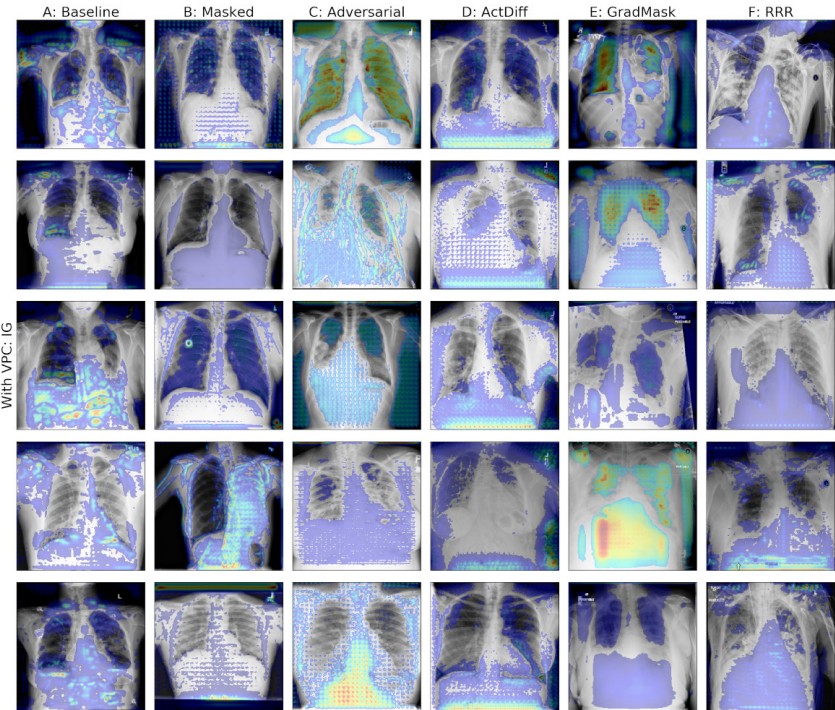

Figure A.17: Randomly-selected images for the RSNA VPC dataset from 5 trained models (1 per row) for each training method, computed using integrated gradients.

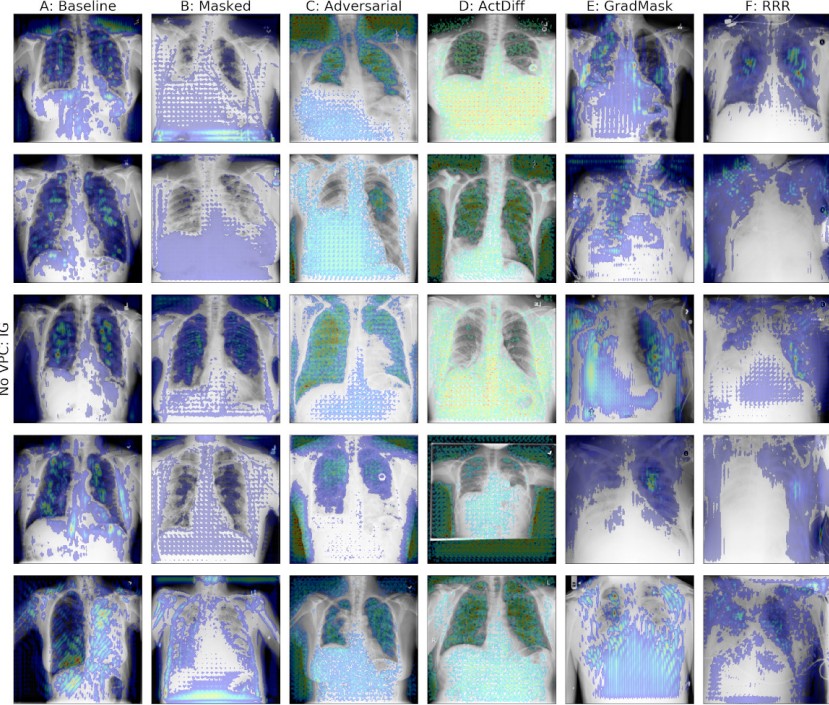

Figure A.18: Randomly-selected images for the RSNA No VPC dataset from 5 trained models (1 per row) for each training method, computed using integrated gradients.

## A.10 ANALYSIS OF MODEL PERFORMANCE WITH EQUIVALENT SKEW IN TRAIN AND VALIDATION SETS

The hyperparameter searches in this paper rely on the existence of a covariate shift between the training and validation distributions: generalization performance to the validation set is used to select the regularization lambdas which encourage the model to ignore the confounder. In the typical machine learning setup, the training and validation set are drawn IID from the same distribution. As we saw on the baseline experiments, where there was no covariate shift in the data and the train/valid distributions were the same, the regularization terms can hurt performance and a hyperparamater search will seek to reduce their impact (since building representations of any confounders will produce good validation performance). This is why in our original experiments it was important that the covariate shift not be equivalent for the train and validation sets, and it is what we recommend in practice (this is easy to accomplish because the exact sources of the covariate shift do not need to be known, so the data can simply be sampled from some different environment where many background variables are likely to differ from the training distribution). Here, we present the results of 5 seeds run for all models on both the X-Ray SPC and RSNA VPC datasets, where the train and validation sets both exhibited a 90/10 skew (exactly matching the skew of the training sets in the original paper), and the test set exhibits a 10/90 skew. We set all hyperparamaters to be the same as found during the search made for the main paper, except for setting the regularization term's lambda to 1 for all methods (where applicable) since this number cannot be searched for because there is no covariate shift in the validation set to determine which lambda works (also see the results in Table 4).

In all cases we observe high validation performance and poor test performance. Interestingly, the method which performs best in both cases is the Masked approach, which has no lambda to tune. For both datasets, it reduces the validation performance in exchange for improved test performance. The gradmask approach also improves test performance in the X-Ray SPC dataset. In total we take these results to suggest that these methods must be used in cases where there exists some known or unknown distribution shift between the training and validation sets to set the correct lambda: they should not both be drawn IID from the same underlying distribution.

| X-Ray: SPC | Experiment | Valid AUC | Test AUC |
|---|---|---|---|
| Baseline | Classification | $0.96 \pm 0.00$ | $0.64 \pm 0.07$ |
| | Masked | $0.93 \pm 0.04$ | $0.78 \pm 0.03$ |
| Domain Invariance | ActDiff | $0.97 \pm 0.00$ | $0.69 \pm 0.03$ |
| | Adversarial | $0.96 \pm 0.01$ | $0.63 \pm 0.03$ |
| Saliency Based | GradMask | $0.98 \pm 0.00$ | $0.74 \pm 0.03$ |
| | RRR | $0.96 \pm 0.01$ | $0.59 \pm 0.06$ |

| RSNA: VPC | Experiment | Valid AUC | Test AUC |
|---|---|---|---|
| Baseline | Classification | $0.92 \pm 0.01$ | $0.37 \pm 0.02$ |
| | Masked | $0.57 \pm 0.12$ | $0.43 \pm 0.06$ |
| Domain Invariance | ActDiff | $0.83 \pm 0.02$ | $0.33 \pm 0.02$ |
| | Adversarial | $0.82 \pm 0.06$ | $0.34 \pm 0.02$ |
| Saliency Based | GradMask | $0.92 \pm 0.00$ | $0.37 \pm 0.01$ |
| | RRR | $0.92 \pm 0.00$ | $0.36 \pm 0.01$ |

Table 5: Mean and standard deviation Valid and Test AUC for all methods tested on the X-Ray SPC and RSNA VPC datasets across 5 seeds.

