# OpenReview forum: "Saliency is a Possible Red Herring When Diagnosing Poor Generalization"
_ICLR.cc/2021/Conference — ICLR 2021 Poster_

### Official Review · AnonReviewer3 · 2020-10-15
**On attributions and generalizability: lack of depth and not clearly grounded assumption**

**Rating:** 7
**Confidence:** 5

**Review:**

Updated recommendation after major changes to the submission: thank you for addressing my comments.

Short summary
----------------------------
The authors investigate the relationship between model generalization under distribution shifts and attribution techniques. They hypothesize that imposing “better” attributions in a model (which they define as being more aligned to a mask selected by domain experts) would increase model generalizability. However, they observe that such constraints hurt the performance under no shift, and do not necessarily lead to increased performance or “better” feature attribution maps under shift.

Strengths
-----------------------------
This work investigates 3 datasets: one synthetic dataset where the effects are well-controlled for, as well as 2 manipulations of real-world data in medical imaging. It includes different and recent techniques to constraint model learning by masked representations, and does not make bold claims about the results.

Weaknesses
----------------------------
There are however a few weaknesses that represent major concerns to me:
1) The main assumption underlying this work is that improving feature attributions will help generalizability. I however do not think that this is the message that was relayed in past publications on the topic (contrarily to what is mentioned in the introduction) and do not think that “better” attributions is sufficient for better generalizability. My reading on this topic is that attribution methods can help in highlighting confounding factors when models do not generalize under distribution shifts. Making this a sufficiency condition is a step that is not well-motivated to me.
2) The reason why I do not believe this assumption is valid is because attribution methods have been shown to not correctly represent model’s decisions and to be sensitive to different factors (including shift-variance, Kindermans et al., 2017, or being susceptible to adversarial perturbations). In addition, different attribution methods will likely display different patterns.
3) Which is why I am wondering whether other attributions, which are more theoretically grounded (e.g. integrated gradients [Sundararajan et al., 2017] or more recent gradient-based techniques like DeepLIFT, or from non-gradients based techniques like SHAP or occlusion) were investigated.
4) The authors mention that masks represent a “good” attribution map. However, there is no guarantee that features in those masks are not affected by distribution shifts. This should be discussed as a limitation of mask-based regularizers.
5) The authors seem to have missed that the “masked” trained classifier also highlights the confounder on the synthetic data, despite it correctly predicting the outputs. I believe this is related to the choice of saliency maps, as “raw” gradients do not “explain” the decision of the model, i.e. it does not display the effect the features have between a bad or neutral decision and the current prediction. For this reason, I would suggest using integrated gradients or another technique.
6) The purpose of the study is quite vague and its results and conclusions lack of depth and reflexion.

Novelty:
------------------------
The authors propose two novel methods to constrain models by using masked representations. While in theory they seem interesting (e.g. activations have been successfully used for OOD detection), the results obtained vary per dataset and there is no clear advantage in terms of model generalization to be using their technique over other methods.
I feel that the main assumption is not well-grounded, and hence, to me, novelty does not reside in the field tackled.

Clarity:
----------------------
Mainly, I found the paper well written and the experiments clear. I would suggest some proof-reading (see minor comments, some repeated or missing words). Mostly I would suggest to revise the introduction as it wasn’t clear to me what the purpose of the study was until well into the experiments. I also found that the introduction is not well matched to the main message of the paper (e.g. it does not mention mask-related constraints).

Rigor:
------------------------
I found that the comparison between the proposed technique and the literature was well explained and sufficient. Hyper-parameters were described and confidence intervals were provided on all results. I enjoyed the fact that three datasets were included. Overall, I think the experiments were well executed.

Detailed comments:
-----------------------------
- “A critical assumption underlying these aforementioned works is that properties of the saliency map are indicative of generalization performance.” I do not agree with that statement given my major concerns (1).
- Intro: masking is mentioned in the key contributions but not before and no justification is used. Such masking also assumes that the shifts across train and test distributions do not impact features in this mask. This seems like a strong hypothesis to me, especially when considering e.g. different imaging sites, or image resolutions.
- It is unclear whether the goal is to obtain models that generalize better or whether it is to obtain feature attribution maps that are consistent with expert input. These two goals can be misaligned, as displayed in the results and should not be conflated.
- actdiff method: can lambda be mentioned in the equation? How would such a regularizer perform in a high dimensional layer (curse of dimensionality)? Why use pre-activation outputs? Were any other versions of this formulation tried?
- How were the masks defined? Is there variability per sample (e.g. different experts) in their definition?
- The authors refer to the synthetic dataset as the changes in the chest x-ray but then have a separate synthetic dataset. The language is confusing and datasets should be defined before being referenced to.
- Synthetic data results: the masked model performs best, ignoring the confounder. However, the saliency map reflects that the attributions are high for the confounder. Therefore, I do not see the same “high correlation” between IoU and AUC that the authors mention. In addition, this, to me, reflects the main limitation of attribution maps: they do not reflect the model’s decision. They rather reflect the local effect of a feature on the label (Lipton, 2016, Ancona et al., 2018). I am wondering why the authors select saliency maps compared to e.g. integrated gradients (Sundararajan et al., 2017) or other gradient-based but more mathematically grounded techniques. Given my own experience of attribution techniques, it is likely that using different attribution methods will lead to different conclusions with respect to the correlation between IoU and AUC.
- I understand the choice in the setup of confounders for both datasets. However I am unsure how this represents real-world settings. For example, training sets might indeed be site-specific, but it would be surprising to me that the test presents the inverse of the confounder. We could for example expect the absence of that confounder (which could be simulated by removing the confounder in the test set for synthetic data), or lower correlations between a feature and the label.
- ActDiff substantially decreases model performance in the absence of confounders
- If the goal is to obtain more generalizable models, other techniques could be envisaged when the relationship between label and confounder is not deterministic, like resampling or reweighting. Were these considered?

Minor:
------------
- Intro: PAC undefined
- intro: the authors conflate the behavior of a model and which inductive biases it relies on, with the obtained saliency maps. This is however a complex and, in my opinion, unanswered question.
- Figure 1: NIH, PC, PA and AP not defined. SPC and VPC are presented succinctly without intuition and we only understand them much later. Maybe this figure should be moved in the results.
- contributions: point 1 needs proof-reading
- “out of distribution feature attribution phenomena”: I searched DeGrave et al for this term but could not find it. If not used elsewhere, I would rephrase as this is a confusing formulation: samples/images can be OOD, and these samples can provide feature attributions, but the feature attributions themselves are not OOD.
- proof-reading of the text is required. Some explanations are poorly framed and can be rephrased (e.g. related works, paragraph 2).
- Zeiler & Fergus, 2013 (published at ECCV 2014) refers to the work on occlusion, where inputs are masked by a baseline value and the changes in predicted risk is defined as their attribution. While this technique provides one attribution per feature, it is not based on the gradients of the network. This paper also does not refer to the term “saliency”.
- Formulas 4 and 5, consistency in the formulations is desirable: if showing for binary classification in most cases, it is better to keep this set up for other formulations (even if they could be extended to multiclass), especially as the experiments are run on binary classification. The limitation of GradMask to binary cases could however be mentioned.
- Figure 5 can be difficult to investigate without colormaps. For instance, it looks to me like the “masked” training model does highlight the confounding factor.

---

> ### Author Response · Authors · 2020-11-15
> **Response to Weaknesses:  We have tried to follow your suggestions in the new version of the paper, uploaded now! We believe this has made the work much stronger.**
>
> We thank the reviewer for all the great points. We have performed new experiments which will be uploaded in a revised document early this week. We wanted to respond to your points now to ensure that these changes have satisfied your concerns with our paper.
>
> > 1. The main assumption underlying this work is that improving feature attributions will help generalizability. I however do not think that this is the message that was relayed in past publications on the topic (contrarily to what is mentioned in the introduction) and do not think that “better” attributions is sufficient for better generalizability. My reading on this topic is that attribution methods can help in highlighting confounding factors when models do not generalize under distribution shifts. Making this a sufficiency condition is a step that is not well-motivated to me.
>
> We shared the belief that saliency could help in highlighting confounding factors. However in this work we demonstrate models which do not indicate incorrect feature attribution yet fail to generalize dramatically (in fact they look at the right area). Therefore our paper is now questioning this line of thinking.
>
> This impression that a good saliency map implies a good prediction is common in applied domains (such as medical) which is the reason we wrote this paper https://www.nature.com/articles/s41598-019-42557-4 https://www.nature.com/articles/s41591-020-0942-0 . We believe the evidence that we present in this paper will serve to help guide those researchers to evaluate models with more rigor.
>
> The idea that correcting models to have "good" saliency would yield good generalization is present in the ML literature as well [RRR, https://arxiv.org/abs/1906.10670].
>
> > 2. The reason why I do not believe this assumption is valid is because attribution methods have been shown to not correctly represent model’s decisions [... &] different attribution methods will likely display different patterns.
>
> Thank you for raising this important point. We maintain that this paper actually supports your claim, and provides experimental evidence in a real-world application that the assumption you describe is not valid. We performed this research to challenge that assumption to have a citable example that this is not valid. Your concern that Input gradients might not be representative of the model’s attributions is a good one, and to address this we have calculated both Integrated Gradients and Occlusion-based attribution maps for all experiments. In the main text, we present the Integrated Gradients images, and show all three methods side-by-side in the appendix. Interestingly, we found all methods gave similar but not identical attribution patterns.
>
> > 3. Which is why I am wondering whether other attributions, which are more theoretically grounded (e.g. integrated gradients [Sundararajan et al., 2017] or more recent gradient-based techniques like DeepLIFT, or from non-gradients based techniques like SHAP or occlusion) were investigated.
>
> We originally focused on input gradient in line with previous work [GradMask, RRR], but have now also computed the integrated gradients and occlusion-based metrics for all experiments and included them in the Appendix Figures A3-7 and A13 for the overview plots (now uploaded). We believe that this paper is now much stronger with these included. Thank you!
>
> + Example Synthetic Images: https://figshare.com/s/84ebabd78fe7d3b0b271
> + Example XRay Images: https://figshare.com/s/d1dc30ca2241853cf219
>
> > 4. The authors mention that masks represent a “good” attribution map. However, there is no guarantee that features in those masks are not affected by distribution shifts. This should be discussed as a limitation of mask-based regularizers.
>
> We agree and have added this in a limitations section.
>
> > 5. The authors seem to have missed that the “masked” trained classifier also highlights the confounder on the synthetic data, despite it correctly predicting the outputs. I believe this is related to the choice of saliency maps, as “raw” gradients do not “explain” the decision of the model [...] I would suggest using integrated gradients or another technique.
>
> While it is true that the masked approach appears to highlight the confounder, it does also do a slightly better job at distributing attribution to the ‘+’ symbols. Interestingly, we see the same behaviour for Input Gradients, Integrated Gradients, and the Occlusion approach. We have added experiments using different methods to generate saliency maps and present them side-by-side in the appendix, showing integrated gradients in the main text.
>
> > 6. The purpose of the study is quite vague and its results and conclusions lack of depth and reflexion.
>
> Aside from adding new experiments as you suggested, we have tried to clarify the purpose of this work, and expand upon the conclusions, by expanding both sections to make the purpose of the work clearer.

---

> > ### Comment · AnonReviewer3 · 2020-11-23
> > **Thank you for the reply**
> >
> > Dear Authors,
> >
> > Thank you for your careful reply and for taking into account most of my comments.
> >
> > I believe the work is now much clearer in terms of the assumption tested, and I will amend my recommendation accordingly.
> >
> > Remaining comments:
> > - Minor proof-reading is required. I understand that quite some text was modified. There are still a few missing words and typos, e.g. "All models we trained using the Adam optimizer (Kingma & Ba, 2014) using early stopping on the validation loss. Hyperparamters... ".
> > - The "masked" model displays higer variance in terms of generalizability. I am wondering whether this wouldn't highlight the presence of the confounder *in the signal included in the mask*, and would be a manifestation of under-specification (D'Amour et al., 2020)? This might be interesting to discuss.
> > - The masked prior is one specific way to enforce domain knowledge. In real-world applications, I believe there are only a few cases where masks represent such prior and where they would be available. This is a general criticism of mask-based techniques, that could be interesting to mention in the limitations (thank you for adding this section).
> > - The current version of the work departs quite significantly from the assumption that enforcing domain knowledge is useful for generalization and/or attribution. I wonder if an extension could be to not focus on mask-based techniques, but illustrating this phenomenon on different types of techniques, including methods defined in the field of robustness or fairness (as mentioned in referring to MMD). This can be left for future work, but a journal paper that quantifies the correlation between AUC and attribution quality (either IoU or other metrics, e.g. see https://github.com/google-research-datasets/bam) would be valuable. Figure 8 is very interesting, but the conclusion that the correlation is "weak" is a bit disappointing.

---

> > > ### Author Response · Authors · 2020-11-25
> > > **Thank you for your careful review and kind assessment of our response!**
> > >
> > > We appreciate you taking the time to review our improvements to the manuscript in response to your comments. Hopefully we have addressed your final points in the last updated version of the PDF (uploaded).
> > >
> > > > Minor proof-reading is required. I understand that quite some text was modified. There are still a few missing words and typos, e.g. "All models we trained using the Adam optimizer (Kingma & Ba, 2014) using early stopping on the validation loss. Hyperparamters... ".
> > >
> > > Thank you again and apologies on this front, we believe we have caught the lion's share of the errors now.
> > >
> > > > The "masked" model displays higer variance in terms of generalizability. I am wondering whether this wouldn't highlight the presence of the confounder in the signal included in the mask, and would be a manifestation of under-specification (D'Amour et al., 2020)? This might be interesting to discuss.
> > >
> > > Underspecification is an interesting angle we hadn't considered, thank you. We have added this idea alongside our discussion of how these methods are not guaranteed to mitigate confounding signals inside of the masks, in the limitations section.
> > >
> > > > The masked prior is one specific way to enforce domain knowledge. In real-world applications, I believe there are only a few cases where masks represent such prior and where they would be available. This is a general criticism of mask-based techniques, that could be interesting to mention in the limitations (thank you for adding this section).
> > >
> > > This is a good point, and we have added it to the limitations section, suggesting the need for future work on algorithms that do not require masks.
> > >
> > > > The current version of the work departs quite significantly from the assumption that enforcing domain knowledge is useful for generalization and/or attribution. I wonder if an extension could be to not focus on mask-based techniques, but illustrating this phenomenon on different types of techniques, including methods defined in the field of robustness or fairness (as mentioned in referring to MMD). This can be left for future work, but a journal paper that quantifies the correlation between AUC and attribution quality (either IoU or other metrics, e.g. see https://github.com/google-research-datasets/bam) would be valuable. Figure 8 is very interesting, but the conclusion that the correlation is "weak" is a bit disappointing.
> > >
> > > We agree that this conclusion is interesting but could be much stronger. However we don't believe our data supports a much stronger conclusion than "the correlation that appears to exist for the synthetic data was not found in the real-life datasets we tested". We sincerely hope that future work will extend this inquiry beyond mask-based algorithms and in such a way that we will have a more definitive answer on the topic. We have indicated the need for future work that does not rely on these masks in our limitations as a prompt for future work.
> > >
> > > We would like to thank you again for your attentive review. We believe the paper is much stronger with your guidance and hope you are satisfied with our changes.

---

> ### Author Response · Authors · 2020-11-15
> **Response to Novelty / Clarity / Rigor**
>
> > The authors propose two novel methods to constrain models by using masked representations. While in theory they seem interesting (e.g. activations have been successfully used for OOD detection), the results obtained vary per dataset and there is no clear advantage in terms of model generalization to be using their technique over other methods. I feel that the main assumption is not well-grounded, and hence, to me, novelty does not reside in the field tackled.
>
> In the presence of a covariate shift, most of the algorithms presented easily outperform the baseline classification model. In the RSNA data, the difference is non-trivial, the baseline classification model only scores an AUC of 0.2, whereas the two domain adaptation-based approaches, ActDiff and Adversarial, score an AUC of 0.68 and 0.62 respectively. In the X-Ray data, the baseline performs much better due to the less severe effect of the confounder (AUC=0.7), but both GradMask and Actdiff improve on this score by a few percentage points.
>
> > Mainly, I found the paper well written and the experiments clear. I would suggest some proof-reading (see minor comments, some repeated or missing words). Mostly I would suggest to revise the introduction as it wasn’t clear to me what the purpose of the study was until well into the experiments. I also found that the introduction is not well matched to the main message of the paper (e.g. it does not mention mask-related constraints).
>
> Thank you for drawing our attention to this crucial limitation, we have greatly expanded the introduction to make the purpose of this work clearer.
>
> >I found that the comparison between the proposed technique and the literature was well explained and sufficient. Hyper-parameters were described and confidence intervals were provided on all results. I enjoyed the fact that three datasets were included. Overall, I think the experiments were well executed.
>
> We appreciate your kind words on this matter!

---

> ### Author Response · Authors · 2020-11-15
> **Response to Detailed Comments.**
>
> > “A critical assumption underlying these aforementioned works is that properties of the saliency map are indicative of generalization performance.” I do not agree with that statement given my major concerns (1).
>
> While there might not be a broad consensus that “properties of the saliency map are indicative of generalization performance”, we have come across multiple works which operate on this sort of assumption, and we wrote this work in response to those papers. We have tried to make the literature we are responding to more explicit in the introduction, as we agree there are many dissenting views on this topic.
>
> > Intro: masking is mentioned in the key contributions but not before and no justification is used. Such masking also assumes that the shifts across train and test distributions do not impact features in this mask. This seems like a strong hypothesis to me, especially when considering e.g. different imaging sites, or image resolutions.
>
> You are correct that this is likely too strong a hypothesis to be completely true. However, in practice, our attribution priors did improve generalization performance, so we have evidence that it is at least partially true (particularly for the RSNA data where the performance gain is over 3x from the baseline). We will rework the introduction to make this assumption more clear, and detail the limitations of this hypothesis.
>
> > It is unclear whether the goal is to obtain models that generalize better or whether it is to obtain feature attribution maps that are consistent with expert input. These two goals can be misaligned, as displayed in the results and should not be conflated.
>
> Based on previous work, we hypothesized that we could improve generalization by improving saliency maps using attribution priors. We were surprised to learn that while attribution priors seem to aid in generalization, they had an inconsistent effect on the attribution maps. We have tried to make this logic more clear in the introduction.
>
> > actdiff method: can lambda be mentioned in the equation? How would such a regularizer perform in a high dimensional layer (curse of dimensionality)? Why use pre-activation outputs? Were any other versions of this formulation tried?
>
> We added a lambda now. We experimented with matching activations at every layer but this was slow, hurt performance if the penalty was applied too close to model input, and found that the good results were achieved with the fastest convergence times if the penalty was only applied in the penultimate layer, which is not of high dimension.  We have explicitly addressed your very good point about using the l2 distance in high dimensional spaces when we introduce the method in the paper. Depending on the use case, one could consider using a l1 norm or something even smaller.
>
> >How were the masks defined? Is there variability per sample (e.g. different experts) in their definition?
>
> The masks were defined by radiologists for the PadChest, NIH, and RSNA datasets. These are standard public datasets for radiology research and are well-documented. Unfortunately, we do not have a measure of inter-rater reliability for these masks, but this is a common issue in radiology e.g., https://www.sciencedirect.com/science/article/abs/pii/S0968016012001135. Given the large nature of the masks used for the RSNA dataset, we do not think that inter-rater reliability of the masks was a crucial flaw of the paper, especially since using the masks improved generalization performance.
>
> > The authors refer to the synthetic dataset as the changes in the chest x-ray but then have a separate synthetic dataset. The language is confusing and datasets should be defined before being referenced to.
>
> Thanks for pointing this out, we have clarified our language.
>
> > Synthetic data results: the masked model performs best, ignoring the confounder. However, the saliency map reflects that the attributions are high for the confounder. Therefore, I do not see the same “high correlation” between IoU and AUC [...] it is likely that using different attribution methods will lead to different conclusions with respect to the correlation between IoU and AUC.
>
> The masked, actdiff, adversarial, and to a lesser degree the grandmask approach, all learn to avoid the confounder. In Figure 8, one can see the relationship between Test IOU and Test AUC for all experiments -- the items in blue are for the synthetic dataset, and there is a clear positive correlation between the IOU and test AUC score for this dataset, with some classes of approaches (mask and domain invariance) outperforming the saliency and baseline approaches. It is true that the masked approach also pays attention to the confounder, but it does not do so exclusively, which might explain the discrepancy.
>
> We completely agree RE: the attribution method, which is why we re-computed all attribution maps using integrated gradients for the results. Thank you very much for this great suggestion!

---

> ### Author Response · Authors · 2020-11-15
> **Further Response to Detailed Comments.**
>
> > I understand the choice in the setup of confounders for both datasets. However I am unsure how this represents real-world settings...
>
> We agree that our datasets represent an exaggerated setting. We anted to see worse than random performance to know that the models are truly using the wrong features instead of simply guessing at random. This is also done in the causal learning literature (i.e, IRM, ILC) to demonstrate extreme overfitting in the face of a covariate shift. In practice, the test set is more likely to be a modulation of the relationship as you suggest. If we have enough time, we will run the SPC / VPC models on the test sets from the non-confounded datasets to evaluate their performance relative to the No-SPC/No-VPC models trained, and notify you that those experiments were run.
>
> > ActDiff substantially decreases model performance in the absence of confounders
>
> This is true and likely due to the nature of our hyperparameter search. If we had allowed the model to search for a smaller lambda, the search would have learned to turn the actdiff penalty down, or completely off, when there exists no covariate shift in the data. Appendix Table 3 shows the results of these searches, where we were able to recover baseline performance in the X-Ray No SPC dataset, but not RSNA No VPC. We suspect this is due to the compute budget we alloted for the search (20 iterations) and the very wide range of lambda values attempted. With more time, we suspect we would be able to resolve this issue with a thorough enough search: this sensitivity to the lambda value might be a downside of the ActDiff approach. It is possible that some tweak to the loss would improve this behaviour, but we don't see this as a crucial issue for the work as our results do not rely on ActDiff being a superior method for controlling feature attribution.
>
> > If the goal is to obtain more generalizable models, other techniques could be envisaged when the relationship between label and confounder is not deterministic, like resampling or reweighting. Were these considered?
>
> This can be done IIF we actually know there is a confounder, but in many real life applications this is both unknown and unknowable. The examples we chose are synthetic in nature meant to represent a worst-case scenario where the effect of these approaches can be easily measured and understood. We agree that it is unlikely this is as bad as it would be in a real-life scenario, but problems of this nature are common in applications.

---

> ### Author Response · Authors · 2020-11-15
> **Response to Minor Comments: Thank you for the excellent feedback on our work. Please See the Updated PDF With New Experiments and Discussion.**
>
> > intro: the authors conflate the behavior of a model and which inductive biases it relies on, with the obtained saliency maps. This is however a complex and, in my opinion, unanswered question.
>
> We have reformulated to be clear we are talking about the constructed features and not the model’s inductive biases.
>
> > Figure 5 can be difficult to investigate without colormaps. For instance, it looks to me like the “masked” training model does highlight the confounding factor.
>
> Yes the masked training model does highlight the confounding factor. We found this method does not control saliency in a meaningful way across all experiments, even though it often aided in generalization.
>
> > Intro: PAC undefined
>
> Thank you, we now define this in the text.
>
> > Figure 1: NIH, PC, PA and AP not defined. SPC and VPC are presented succinctly without intuition and we only understand them much later. Maybe this figure should be moved in the results.
>
> We agree and have moved this figure to the experimental protocol section.
>
> > contributions: point 1 needs proof-reading
>
> Thank you for pointing this out!
>
> > “out of distribution feature attribution phenomena”: I searched DeGrave et al for this term but could not find it. If not used elsewhere, I would rephrase as this is a confusing formulation: samples/images can be OOD, and these samples can provide feature attributions, but the feature attributions themselves are not OOD.
>
> Thank you for pointing out this unclear language, which we have reworked.
>
> > proof-reading of the text is required. Some explanations are poorly framed and can be rephrased (e.g. related works, paragraph 2).
>
> Thank you for your feedback, we have done a thorough proof read of the document and specifically addressed related works
> paragraph 2.
>
> > Zeiler & Fergus, 2013 (published at ECCV 2014) refers to the work on occlusion, where inputs are masked by a baseline value and the changes in predicted risk is defined as their attribution. While this technique provides one attribution per feature, it is not based on the gradients of the network. This paper also does not refer to the term “saliency”.
>
> Thank you for pointing out this mistake, as we have now implemented the occlusion method, we have correctly cited this work (the ECCV 2014 version) in the appropriate section when we describe the method, and we have corrected the language.
>
> > Formulas 4 and 5, consistency in the formulations is desirable: if showing for binary classification in most cases, it is better to keep this set up for other formulations (even if they could be extended to multiclass), especially as the experiments are run on binary classification. The limitation of GradMask to binary cases could however be mentioned.
>
> Thank you, we have modified the formulations to be consistent for the binary classification case, noting when the equation can be naturally extended to multi-class.
>
> Thank you for your excellent feedback on our work.

---

### Official Review · AnonReviewer4 · 2020-10-28
**Review for Saliency is a Possible Red Herring When Diagnosing Poor Generalization**

**Rating:** 7
**Confidence:** 4

**Review:**

The reviewed paper explores the relationship between the quality and spatial distribution of the saliency maps produced at inference time and the model's generalization performance. The authors employed a number of existing methods as well as proposed and implemented their own technique (ActDiff) to align saliency maps with causally plausible regions. All methods were applied on synthetic and real-world data in a series of clever experiments, showing little correlation between saliency map spatial alignment and performance on unseen data.

Overall, the paper seems to be technically sound, claims justified, and supported by the evidence presented in tables and figures.  Importantly, the paper raises a very interesting point, challenging the status quo in the field. The manuscript is relatively easy to read and understand. For all these reasons, I vote for accepting.

Major questions-concerns:
* Figures 6 and 7 show mean saliency maps from randomly selected test images. Columns of these images show outputs from different algorithms used in this work. My question is why do saliency maps produced by methods such as Masked on test images show significant activity in the regions that were explicitly made useless for training (by randomly shuffling pixels outside masks)? I believe this requires a more elaborate and explicit answer.
* Perhaps it makes sense to openly recognize that before completely dismissing the validity of using saliency maps for diagnosing overfitting a lot more datasets must be studied as the two presented in the paper, may not be enough to provide conclusive evidence.

Several minor comments and questions:
* Some acronyms, e.g. PAC on the first page is not defined before being used.
* Caption for Figure 1 can be made more clear, as there is no explanation of what "pathology correlation with site/view" means. It is only later in the text, the authors add that they have intentionally biased positive cases by sampling them mostly from either one site or one view. But before reading this part, the caption remains confusing for the reader.
* It might be a good idea to reformulate contributions of the paper into nouns instead of verbs e.g. instead of "Create a dataset" - "A dataset".
* A sentence from the related work section, namely: "Zhuang et al. (2019) was additionally designed..." should be reformulated.

Updates: Thanks for the authors' response. I believe this paper is valuable for the field and community and therefore I recommend this paper to be accepted.

---

> ### Author Response · Authors · 2020-11-18
> **Thank You For Your Comments Regarding Our Work! Please See the Updated PDF**
>
> > Figures 6 and 7 show mean saliency maps from randomly selected test images. Columns of these images show outputs from different algorithms used in this work. My question is why do saliency maps produced by methods such as Masked on test images show significant activity in the regions that were explicitly made useless for training (by randomly shuffling pixels outside masks)? I believe this requires a more elaborate and explicit answer.
>
> Thank you for pointing this out. We have updated the text to explain what we believe to be going on here. Briefly, since the model is convolutional, during training, the model learns to build features from within the masks. During testing, any texture or shape learned from inside of the mask might also match local regions of the image outside of the mask. There is no restriction on the regions of the image that can be predicted from during test time. This is a failure of less sophisticated methods like masking.
>
> > Perhaps it makes sense to openly recognize that before completely dismissing the validity of using saliency maps for diagnosing overfitting a lot more datasets must be studied as the two presented in the paper, may not be enough to provide conclusive evidence.
>
> Yes we agree that these datasets are not sufficient to make a conclusive claim broadly, and we hope this work encouraged future research. We will make this limitation more explicit in the conclusions with a dedicated paragraph.
>
> > Some acronyms, e.g. PAC on the first page is not defined before being used.
>
> Thanks, we have fixed this.
>
> > Caption for Figure 1 can be made more clear, as there is no explanation of what "pathology correlation with site/view" means. It is only later in the text, the authors add that they have intentionally biased positive cases by sampling them mostly from either one site or one view. But before reading this part, the caption remains confusing for the reader.
>
> This figure has been moved to the experimental section of the article, which we think makes it easier to understand.
>
> > It might be a good idea to reformulate contributions of the paper into nouns instead of verbs e.g. instead of "Create a dataset" - "A dataset".
>
> Thanks, good idea.
>
> > A sentence from the related work section, namely: "Zhuang et al. (2019) was additionally designed..." should be reformulated.
>
> Thanks, we have done this.

---

> > ### Comment · AnonReviewer4 · 2020-11-22
> > **Thank you for the reply**
> >
> > Thank you for reflecting on my questions.

---

### Official Review · AnonReviewer1 · 2020-10-28
**Interesting approach to a (very) real albeit (very) specific problem**

**Rating:** 7
**Confidence:** 3

**Review:**

This paper addresses the potential correlation between saliency map values and model generalization ability in image analysis with a focus on medical image.
The paper falls well within the scope of the conference.
It is overall well written, but it is so crammed with information that, in order to comply with paper length limits, its 'digestion' and
interpretation becomes difficult at times (this is clear in the case of images, whose meaning often has to be half-guessed due to lack of details)
The problem this study deals with is definitely important in the context of medical decision making on the basis of image, and the problems it pinpoints are more than real (small sample sizes, inter-site heterogeneity, bias due to human intervention in the masking process, etc)
The proposal seems sound to me and the experiments convincing. My main qualm concerns their specificity, given that they address a very specific domain.

---

> ### Author Response · Authors · 2020-11-18
> **Thanks for Your Kind Assessment, Please See the Updated PDF with New Experiments and Text**
>
> > This paper addresses the potential correlation between saliency map values and model generalization ability in image analysis with a focus on medical image. The paper falls well within the scope of the conference. It is overall well written, but it is so crammed with information that, in order to comply with paper length limits, its 'digestion' and interpretation becomes difficult at times (this is clear in the case of images, whose meaning often has to be half-guessed due to lack of details)
>
> Thank you for your kind assessment of our manuscript! We will take your suggestion to clarify the text as much as possible seriously. In particular, we will take advantage of the appendix to flesh out experimental details that are briefly described in the main text.
>
> > The problem this study deals with is definitely important in the context of medical decision making on the basis of image, and the problems it pinpoints are more than real (small sample sizes, inter-site heterogeneity, bias due to human intervention in the masking process, etc) The proposal seems sound to me and the experiments convincing. My main qualm concerns their specificity, given that they address a very specific domain.
>
> While we understand that we tackled the problem of medical imaging generalization specifically in this paper, the methods and problems addressed are applicable to any imaging application where there might exist a covariate shift between the training and test distributions, and our masking approaches could be used in those contexts. We hope the reader focuses less on the specifics of our masking approach, however, and focus more on the problem of relying on these attribution methods to diagnose model correctness. We have expanded the text to focus the attention on these findings.

---

### Official Review · AnonReviewer2 · 2020-10-31
**Official Blind Review #2**

**Rating:** 6
**Confidence:** 4

**Review:**

--- Summary ---

This paper focuses on the confounder problem that spatially-seperated image regions (e.g. shoulders of xray images) might spuriously correlated with the target (e.g. pneumonia). If given a human-labeled region that is deemed important, we can decrease this spuriousness by regularizing the model toward the important region. They not only compare with several existing saliency-based methods (RRR and GradMask), but also propose 2 new methods (ActDiff, Adversarial) inspired from domain adapataion literature that the representation of the classifier should be similar between original image and the masked image (the image that the non-important region is shuffled). They compare in 1 synthetic dataset and 2 xray datasets. They show that (1) these methods (sometimes) hurt generalization when spuriousness does not exist, and (2) the model's saliency map is only weakly correlated with generalization performance, and thus doubting the validtiy of using saliency maps for diagnosing whether a model is overfit to spurious features.

--- Pros ---

1. Very detailed hyperparameter search and lots of repeated run (10) to get good standard deviation
2. Visualizations of average test images are very intriguing.

--- Major comments ---

1. It seems that when there is covariate shift, the ActDiff performs the best but hurts the performance when no shift exists, but saliency-based methods do not suffer this. But given that you can tune the lambda of the regularization, why could it happen? Can't you just pick lambda close to 0?

2. Some visualization looks suspiciously abnormal. For example, RRR in Figure 5 has a big blur in the middle. Why is that? Also, in Figure 7, Masked focus on the neck to predict but neck is clearly outside of the bounding box. How should we explain this phenomenon? Besides, in Figure 7 upper rows (VPC), Adversarial seems to have better masks by focusing on the lungs, and ActDiff does not. But their IOU is reversed: Adversarial has lower IOU and ActDiff has higher IOU in Table 2.

3. In all the experiments, the validation set is always assumed correct without any background shift. While in real scenario it might not always be easy to access to a validation dataset without any shift. When we instead only have a biased validation dataset, which method will perform better? Will the result change?

--- Minor Comments ---

1. Some hyperparameters for visualization like Gaussian smoothing and thersholding might need justifications. Especially the maximum value capped in 50th percentile seems to be a bit excess.

2. When visualizing the gradients (Fig. 5, 6, 7), maybe we should only include images that model predicts correctly? The gradient of extremely wrong predictions might not be very meaningful.

3. The Figure 8 shows the scatter plot between Test AUC and IOU. It seems Synthetic and Xray SPC have much higher IOU overall. Maybe because their bounding box or confounder is location-wise fixed while only the RSNA VPC has varying bounding boxes?

4. The masked baseline perform much worse in No VPC with AUC=0.5 which is random guessing. But it does not happen in other datasets. Maybe it's because the VPC has smaller bounding box and thus only access to such region is too difficult?

5. The lambda should be included in all the equations (eq. 1 to 5).

6. The final hyperparameter should be reported in the appendix.

--- Overall evaluations ---

Overall I like this paper. The hyperparameter tuning is very thorough. The conclusion is good that great saliency map does not mean better accuracy and vice versa. The experimental results are a bit unsatisfying that no real data is improved. And sometimes the simple masked baseline outperform others. I am happy to increase my score if my major concerns are addressed.

---

> ### Author Response · Authors · 2020-11-18
> **Response to Minor Comments**
>
> > 1. Some hyperparameters for visualization like Gaussian smoothing and thersholding might need justifications. Especially the maximum value capped in 50th percentile seems to be a bit excess.
>
> The gaussian sigma of 1 is small, especially for images with 224x224 resolution, and was done primarily to remove speckly noise from the attribution maps, which distract from the underlying pattern. The threshold was a more arbitrary choice, but we thought the 50th percentile would be easier for the reader to visualize: without this threshold, the anatomy is harder to see. For the values in the tables, we binarized at a percentile that matches the size of the supplied mask, not the 50th percentile, so the qualitative decision made for the figures does not impact the interpretation of the numbers in the tables.
>
> > 2. When visualizing the gradients (Fig. 5, 6, 7), maybe we should only include images that model predicts correctly? The gradient of extremely wrong predictions might not be very meaningful.
>
> Thank you for this helpful suggestion. We have generated figures for samples where A) the model is correct, and B) the model is incorrect on the test set, to determine how different these attributions are. These plots are found in the appendix A6, computed using Integrated Gradients.
>
> > 3. The Figure 8 shows the scatter plot between Test AUC and IOU. It seems Synthetic and Xray SPC have much higher IOU overall. Maybe because their bounding box or confounder is location-wise fixed while only the RSNA VPC has varying bounding boxes?
>
> Yes, you are correct. For the synthetic dataset, the task is extremely easy, so a high IOU isn’t surprising. The saliency maps are quite precise (see Appendix A9). Meanwhile for XRay SPC, the task is simply to predict away from the image border: one can imagine that many saliency maps that are simply focused away from the shoulder will perform well in this case. In contrast, we evaluated the RSNA images by their overlap with bounding boxes around the disease, which is variable across images. This is a harder task for the model to localize well.
>
> > 4. The masked baseline perform much worse in No VPC with AUC=0.5 which is random guessing. But it does not happen in other datasets. Maybe it's because the VPC has smaller bounding box and thus only access to such region is too difficult?
>
> This is reasonable and likely correct although we can’t know for sure. The masked task for the synthetic dataset is quite easy: the network only needs to learn two shapes (+, -) and with masked dataset, it learns how to count the + symbols during training. With the XRay dataset, the gross anatomy of the entire lung is available including the context around. Meanwhile, for the RSNA data, the model is only able to consistently see what is inside the lung, without context. The model is likely learning image textures that might appear elsewhere in the image, leading to bad generalization.
>
> > 5. The lambda should be included in all the equations (eq. 1 to 5).
>
> We have done this in the updated PDF.
>
> > 6. The final hyperparameter should be reported in the appendix.
>
> We have done this in the Appendix.
>
> > Overall I like this paper. The hyperparameter tuning is very thorough. The conclusion is good that great saliency map does not mean better accuracy and vice versa. The experimental results are a bit unsatisfying that no real data is improved. And sometimes the simple masked baseline outperform others. I am happy to increase my score if my major concerns are addressed.
>
> Thank you for your thoughtful review. We would like to point out the following: where there exists a covariate shift between the training and test sets, the domain invariance approaches we introduce improve over the baseline for that experiment, which we believe is a valuable contribution. It is true that in the absence of a covariate shift, these methods do not help the model, but we also would not expect them to as the model would not be as likely to learn to use features that are only predictive in the training set distribution (since the test set distribution is the same). Furthermore, we were equally surprised that the masked baseline performed so well on the X-Ray dataset, but perhaps this is less surprising because the masked region is far from the anatomy that determines the class, and it is also worth noting the high variance in performance of that method across seeds. It would be therefore hard to recommend the masked approach be used on datasets, especially when the masks are smaller (note the far worse performance of the RSNA experiments). We have highlighted this in the text.

---

> > ### Comment · AnonReviewer2 · 2020-11-20
> > **Thank you for the reply**
> >
> > Thank you my comments are addressed.

---

> ### Author Response · Authors · 2020-11-18
> **Response to Major Comments -- Please See Updated PDF**
>
> We thank you for these great comments. We have incorporated them to improve the paper. We have added new experiments to the paper based on your comments and discussed them inline below.
>
> > 1. [...] Can't you just pick lambda close to 0?
>
> Thank you for pointing this out. During our experiments, we did a hyperparamater search in the range of 10e-4 - 10 for the different method’s lambdas, never allowing the value to fall to 0. It is true that, if the model was permitted to select much smaller values as part of the search, we might have obtained better performance when there exists no covariate shift between the training and validation sets. If it had selected zero, we would anticipate the model would perform the same as the baseline model, as the actdiff loss term is the only difference between these models.
>
> To verify this, we performed a hyperparameter searches for all actdiff experiments allowing the lambda to be as small as 10^-16, and followed this with a 10-seed run to verify the results. For the X-Ray No SPC dataset, we recovered baseline performance with a very small lambda, but we did not have the same luck with the the RSNA No VPC dataset, although performance did improve slightly (AUC=0.61). We have seen anecdotal evidence that the model can perform better with an incredibly small search range (10^-32), achieving a test performance of AUC=0.66, but this performance does not get us very close to the baseline performance of AUC=0.76. We therefore conclude that, even with incredibly small lambdas, actdiff can hurt performance relative to baseline with some, but not all datasets. We speculate that this is simply due to the masks being smaller on the RSNA dataset, making this problem harder, and a tweak to the loss function might resolve this problem, but we wanted to present you with the results as we have them now. These findings are detailed in the Appendix of the paper (Section A.8).
>
> > 2. Some visualization looks suspiciously abnormal.
>
> It is true that the saliency maps were sometimes unusual. In the case of  Figure 5 (the synthetic dataset), RRR had trouble finding a good solution on this dataset (a problem not seen with real data). In response to reviewer 4, we have reproduced these results using two alternative attribution methods: integrated gradients and occlusion, which in the case of RRR, give a more reasonable attribution (low values everywhere other than the confounder). We present all methods in the appendix and have replaced all saliency maps in the paper with the integrated gradients approach, which is more theoretically sound and more likely to elucidate the features which drove the prediction.
>
> In Figure 7, it is also true that the model is using the neck, or some edge feature, to make the prediction. This finding is even more apparent in the new attribution maps produced for reviewer 4. With a CNN, simply restricting the input space during training does not prevent the model from learning a feature from within the mask that shares properties with regions outside of the mask, which can be exploited during test time when the full image is presented. We now highlight this in a new limitations section near the end of the paper (section 5).
>
> We agree that the IOU scores do not always line up completely with what one would expect intuitively from the saliency maps. This might be because the segmentations are generated from saliency maps via a threshold: the number of pixels in the mask is counted, and the top % of the saliency map with the matching number of pixels as the mask is binarized to calculate the IOU. In many images, you can see that the maximum intensities (red) sometimes fall outside of the lungs, e.g., the shoulder, even though the relevant anatomy is also highlighted. This could lead to some of the binaized saliency ending up outside of the mask. We’re not sure of a better evaluation metric: if we use a fixed threshold for all images, the evaluation would not be fair as the mask sizes vary a lot between the samples in the RSNA dataset.
>
> To show you exactly what is going on, we have produced a set of randomly-sampled saliency maps (using integrated gradients as per Reviewer 4) across 5 of the seeds for all experiments in the Appendix section A.9. In the real datasets, many interesting phenomena are visible in these images, and many of the models that generalize well appear to output rather artefactual saliency maps in some cases, but we view this as a support for our central argument in the paper.
>
> > 3. [...] the validation set is always assumed correct without any background shift[?]
>
> The validation set has the reverse background shift for all VPC/SPC experiments so that we can see overfitting (as we did for RSNA): a model performing worse than chance. These methods will work if the validation set has a different unknown shift than the training set. We do not require a non-shifted validation set, the only requirement of these approaches are input masks.

---

> > ### Comment · AnonReviewer2 · 2020-11-20
> > **Thank you for the reply**
> >
> >
> > Q1 and Q2 are addressed. Thank you.
> >
> > For Q3, sorry for the confusion. I should rephrase it to "the validation set has the same i.i.d. distribution from the test set". So in your experiments the validation set has the same bias as test set. And in real-world applications having a "clean" validation set (here clean means i.i.d. from the intended test distribution) is not very easy or non-trivial.

---

> > > ### Author Response · Authors · 2020-11-22
> > > **Question**
> > >
> > > For Q3 how should the splits be defined for us to run new experiments:
> > >
> > > A) Train 90/10, valid 10/90, test 50/50
> > >
> > > B) Train 90/10, valid 90/10, test 10/90
> > >
> > > or something else?

---

> > > > ### Comment · AnonReviewer2 · 2020-11-23
> > > > **Just a thought**
> > > >
> > > > I was thinking more like B) or
> > > > C) Train 90/10, valid 90/10, test 50/50.

---

> > > > > ### Author Response · Authors · 2020-11-25
> > > > > **Brief Experimental Results Added in Appendix A 10 (See Updated PDF)**
> > > > >
> > > > > You raise an interesting point: the traditional approach for constructing a train and validation set would be to sample IID from some common underlying data distribution when generating the splits, whereas we explicitly introduce a covariate shift *and then use the validation set to select our hyperparamaters*.
> > > > >
> > > > > This is actually crucial, because as we saw before with the actdiff experiments, in the presence of no shift, many of these methods prefer to turn regularization off: if the confounder is useful in the training set, it would also be useful in the validation set, in this setup.
> > > > >
> > > > > In light of this, we ran 5 seeds per model on the VPC/SPC datasets, where the train and validation sets both exhibited a 90/10 skew, and the test set exhibited a 10/90 skew. We could not search for hyperparamaters for the reasons mentioned above, so we set all lambdas to 1.
> > > > >
> > > > > Most methods did not do a good job of preventing overfitting: validation AUCs were far higher than test AUCs. The one method that performed best was the Masked condition, which is maybe unsuprising because it does not have any hyperparamaters to tune. In this case, we observed the expected behaviour: a drop in Valid AUC performance accompanied by an increase in Test AUC performance.
> > > > >
> > > > > We take these results as supporting evidence that our experimental protocol for the paper was correct. In practice, finding two train and validation datasets with covariate shifts is quite easy: it can be a second dataset collected under some different conditions, using a different instrument, or of a different population. The key is that the source of the confounding does not need to be known, but rather, there should be some shift in the prevalence of the confounding variable between the train and validation set. Therefore, there is an advantage to the validation set being more different than the training set, in contrast to the standard ML setup.
> > > > >
> > > > > Please see Appendix A 10 where we include a short discussion on this topic and the experimental results.

---

### Decision · Program_Chairs · 2021-01-07
**Final Decision**

**Decision:**

Accept (Poster)

**Comment:**

The reviewers all agreed on accepting this paper, stating that it makes a compelling point about the usefulness of saliency methods to diagnose generalization.  The reviewers found that the experiments were a strong point and applauded the thorough hyperparameter tuning and re-runs for statistical significance.  One reviewer commented that the paper was too dense with information, so much so as to make it difficult to digest.  However, overall this seems like an interesting paper that is relevant to the community and will hopefully foster some good discussion about the shortcomings and future directions of saliency methods.